# Enjoy Your Layer Normalization with the Computation Efficiency of RMSNorm

## Abstract

Layer normalization (LN) is a milestone technique in deep learning and has been widely used in various network architectures. It performs centering and scaling over the layer activations of a neural network for each example, stabilizing and accelerating the training of neural network. However, it introduces extra computation cost during inference and the computation problem has recently been addressed by its counterpart RMSNorm that only adopts scaling. This paper investigates how to exploit the theoretical advantages of LN but with the cost of RMSNorm. This paper formally defines the condition that the centering operation of LN can be removed and this condition can be obtained by imposing the column centering constraint on the adjacent linear module before the LN. We propose column centered weight transformation (CCWT) to ensure an LN without centering operation (i.e., RMSNorm) have the same output as the original one in a pre-trained model. Our method can be directly applied to various pre-trained large language models (LLMs) and large vision language models (VLMs) with LN, enabling an immediate reduction in computation cost meanwhile maintaining equivalent prediction during inference. We further propose a reparameterization method, called column based weight centering (CBWC), to ensure the linear module column centered during training. We show that RMSNorm combining CBWC can obtain an equivalent effects to the LN counterpart during training, but with more efficient computation.

## 1 Introduction

Normalization techniques are extensively used in deep neural networks (DNNs) for stabilizing and accelerating the training (Huang et al., 2023). As a seminar work, Batch Normalization (BN) (Ioffe & Szegedy, 2015) improves DNNs' training stability and optimization efficiency by standardizing (centering and scaling) the activations of intermediate DNN layers within a mini-batch of data during training. It uses the population statistics for normalization during inference and this operation can be folded into the adjacent linear layers (Jacob et al., 2018), avoiding the introduction of additional computation cost during inference. In spite of many merits, BN also suffers from the train-inference inconsistent problem, leading to significantly degenerated performance under the scenarios of small-batch size training and domain shifted distributions (Huang et al., 2023). Layer normalization (LN) (Ba et al., 2016) addresses the train-inference inconsistency problem of BN and standardizes the layer input within the neurons for each sample. It has become the key component of Transformer (Vaswani et al., 2017) and its variants (Dai et al., 2019; Xiong et al., 2020; Dosovitskiy et al., 2021), spreading from the Natural Language Processing (NLP) (Radford et al., 2018; Devlin et al., 2019; Raffel et al., 2020) to Computer Vision (CV) (Dosovitskiy et al., 2021; Carion et al., 2020; Cheng et al., 2022) communities. LN has got its firm position (Huang et al., 2023) in the evolution of neural architectures and is currently a basic layer in most of the foundation models (Brown et al., 2020; Alayrac et al., 2022; Kirillov et al., 2023). However, it has to perform the additional standardization during inference, which introduces significant computational cost.

To addresses the computational issue of LN, RMSNorm (Zhang & Sennrich, 2019) is proposed to perform scaling-only operation and is reported to reduce the running time of LN by $7\% \sim 64\%$ on different models, according to the experiments in (Zhang & Sennrich, 2019). Despite its great potential in practice for computational efficiency and wide application in various architectures (Zhang et al., 2024; Team et al., 2024; Mehta et al., 2024), RMSNorm is likely to miss the theoretical merits of centering operation in improving conditioning, which is widely investigated in previous

work (LeCun et al., 1990; Schraudolph, 1998; Montavon & Müller, 2012; Huang et al., 2017). This raises a question that how we can exploit the theoretical advantages of LN but with the computational cost of RMSNorm.

This paper first formally defines the condition that the centering operation of LN can be removed. The condition requires that the input to LN is zero centered of neurons for each sample, so that the removal of centering in LN does not affects the functionality of the network. This condition can be obtained by imposing the column centering constraint on the adjacent linear modules before the LN.

We show that we can satisfy the condition by performing a simple column centered weight transformation (CCWT) for a pre-trained model during inference. This method can ensure an LN without centering operation (*i.e.*, RMSNorm) have the same output as the original one in a pre-trained model. We provide a general method to check whether the centering operation of LN can be removed in a network. We show most of LNs in currently widely used architectures can remove the centering operation, which provides a straightforward benefit in reducing the computation cost during inference. This solution can be directly applied to various pre-trained large language models (LLMs) and large vision language models (VLMs) with LN, enabling an immediate reduction in computation cost without affecting the predictions.

We further propose a reparameterization method, called column based weight centering (CBWC), to ensure the linear module column centered during training. We show that '*CBWC+RMSNorm*' obtains an equivalent effects to the original LN counterpart during training, but with more efficient computation. A network with '*CBWC+RMSNorm*' have equivalent training dynamics to the network with LN, if the LN satisfies the condition that centering operation can be removed. We also conducted experiments to show the effectiveness of '*CBWC+RMSNorm*' when replacing LN, even though the LN can not satisfy the condition.

## 2 NOTATION AND PRELIMINARY

We use $x \in \mathbb{R}$, $\mathbf{x} \in \mathbb{R}^d$ and $\boldsymbol{X} \in \mathbb{R}^{m \times d}$ to denote scalar, vector and matrix respectively, where $\mathbb{R}$ refers to the set of real numbers, and $m, d$ are positive integers. $\mathbf{1}_d$ stands for a $d$-dimension all-one column vector.

**Neural Network.** A neural network can be represented as a function $f(\mathbf{x}; \theta)$, where $\mathbf{x}$ is the input and $\theta$ is the set of all learnable parameters. Take an $L$-layers multilayer perceptron (MLP) as an example, $f_\theta(\mathbf{x})$ consists of stacked linear and nonlinear layers as follows:

$$\mathbf{h}^l = \boldsymbol{W}^l \mathbf{x}^{l-1}, \tag{1}$$

$$\mathbf{x}^l = \varphi(\mathbf{h}^l), \ l = 1, \ldots, L, \tag{2}$$

where the input $\mathbf{x} = \mathbf{x}^0$, the output $f(\mathbf{x}; \theta) = \mathbf{h}^L = \mathbf{x}^L$ and the learnable parameters $\theta = \{\boldsymbol{W}^l, l = 1, ..., L\}$[1]. For each layer, $d_l$ indicates the number of neurons in the $l$-th layer. We have pre-activation $\mathbf{h}^l \in \mathbb{R}^{d_l}$ and the activation $\mathbf{x}^l \in \mathbb{R}^{d_l}$.

**Layer Normalization.** Layer normalization is a basic module in modern DNNs. For a certain layer input $\mathbf{x} = [x_1, x_2, \ldots, x_d]^\top \in \mathbb{R}^d$, LN standardizes $\mathbf{x}$ among the $d$ neurons by performing centering and scaling as:[2]

$$\text{Centering:} \ \tilde{x}_j = x_j - \mu, \quad j = 1, 2, \ldots, d, \tag{3}$$

$$\text{Scaling:} \ \hat{x}_j = \frac{\tilde{x}_j}{\sqrt{\sigma^2 + \epsilon}}, \quad j = 1, 2, \ldots, d, \tag{4}$$

where $\mu = \frac{1}{d} \sum_{i=1}^{d} x_j$ is the mean of $\boldsymbol{x}$ and $\sigma^2 = \frac{1}{d} \sum_{i=1}^{d} \tilde{x}_j^2$ is the second-order moment of $\tilde{\boldsymbol{x}}$. *Centering* ensures zero-mean property among neurons of the input, while *scaling* ensures unit second-order moment property among input elements. LN is usually placed after the linear layer, *i.e.*, normalizing the pre-activation in Eqn. 1.

---

[1]We omitted bias for simplicity, please refer to Appendix A.2 for more details.

[2]In practice, LN have an extra learnable affine transformation after standardization, which we omit here for simplification. Here, $\epsilon$ is a parameter which prevents the denominator from becoming 0.

**RMSNorm**    To reduce the computational usage of layer normalization, Zhang & Sennrich (2019) introduced RMSNorm, with only scaling. RMSNorm is equivalent to scaling—it regard the input as $\tilde{x}$, and get $\hat{x}$ by Eqn.4 directly.

Layer normalization is widely used and achieves excellent performance, but a main problem is its high computational usage. According to the similarity of RMSNorm and LN, we aim to address the issue of high computational cost associated with LN, by replacing it with RMSNorm. However, simple replacements can have potential risks, with decline in performance and adverse effect in training dynamic. In this paper, we first introduce a framework in removing the centering of LN (Section 3). We then discuss the conditions and results in safely replacing LN with RMSNorm in inference (Section 4) and training (Section 5).

## 3    A FRAMEWORK IN REMOVING THE CENTERING OF LN

In this section, we turn to find a way to simplify LN with the computation efficiency of RMSNorm, but with an equivalent performance.We first introduce redundant centering as the condition of equivalent performance. We then propose column centered constraints for linear modules to ensure redundant centering.

### 3.1    REDUNDANT CENTERING

Apparently, we can change LN into RMSNorm, if RMSNorm is capable of achieving equivalent results. Intuitively, with a zero-mean input, RMSNorm will have the same output with LN. Under this situation, we can consider that RMSNorm acts as a scaling operation and, thus, LN has a centering with no effect. Here, we define *redundant centering*.

**Definition 1.** *(Redundant Centering in LN.) For any module $f(\mathbf{x}; \boldsymbol{\theta})$ and an LN directly connected to it, where $\mathbf{x}$ and $\boldsymbol{\theta} \in \Theta$ refers to the input and parameter respectively. We define the centering operation in this LN is redundant, if*

$$RMSNorm(f(\mathbf{x}; \boldsymbol{\theta})) = LN(f(\mathbf{x}; \boldsymbol{\theta})), \forall \mathbf{x}. \tag{5}$$

In other words, if we accomplish the effect of a centering operation in a layer normalization in the module before, we denote that this centering operation is redundant. Therefore, satisfying the condition of a redundant centering, we can delete centering by using RMSNorm in place of LN, and reduce the computation usage.

We thus delve into into a methodology to establish a redundant centering operation.

### 3.2    COLUMN CENTERED CONSTRAINT AND ZERO-MEAN PROPERTY

According to the definition, a redundant centering is independent with the input data. Under this idea, we propose to impose constraint onto the parameter $\boldsymbol{\theta} \in \Theta$. By selecting constraints that endow $\boldsymbol{\theta}$ with a particular property, we aim to realize the centering effect before LN.

In practical neural networks, we divide the whole parameter space $\Theta$ into different subspaces, which parameterize different modules. *These modules are the basic components of the neural network. We can classify the modules into linear modules and non-linear ones based on the transformation it applies on samples.* Therefore, we consider to impose the constrains on the parameters of single modules respectively. Since most parameters lie in the linear modules, we propose *column centered constraint* for the linear modules. Here, we take the linear layer as an example.

**Definition 2.** *(Column Centered Constraint on Linear Layers.) A weight matrix $\boldsymbol{W}_0 \in \mathbb{R}^{d_l \times d_{l-1}}$ is under the column centered constraint, if $\boldsymbol{W}_0$ satisfies*

$$\boldsymbol{W}_0 \in \Gamma_{mlp} = \left\{ \boldsymbol{W} : \sum_{i=1}^{d_l} w_{i,j} = 0, \ j = 1, 2, \ldots, d_{l-1} \right\}, \tag{6}$$

*namely, the mean of all the weights $w_{i,j}, \ i = 1, \ldots, d_l$ for every input $x_j$ is zero.*

We thus aim to demonstrate that the imposition of this particular constraint renders the centering operation redundant. In the following, we will show that the column centered constraint on a linear layer can obtain zero-mean output, achieving the effect of centering operation in a subsequent LN.

**Proposition 1.** *(Zero-mean Property of Column Centered Constraint.) Given a linear layer, if the weight matrix is under column centered constraint, as shown in Definition 2, we figure out that its output is zero-mean.*

*Proof.* Given a certain input $\mathbf{x} \in \mathbb{R}^{d_{l-1}}$ and the output of linear layer $\mathbf{h} \in \mathbb{R}^{d_l}$. In this linear layer, we have $\mathbf{h} = \boldsymbol{W}\mathbf{x}$. Under the constrain of Eqn.6, we have the mean of output $\mathbf{h}$

$$\mu_h = \frac{1}{d_l} \sum_{i=1}^{d_l} h_i = \frac{1}{d_l} \sum_{i=1}^{d_l} \sum_{j=1}^{d_{l-1}} w_{i,j}\, x_j = \frac{1}{d_l} \sum_{j=1}^{d_{l-1}} \left( \sum_{i=1}^{d_l} w_{i,j} \right) x_j = \frac{1}{d_l} \sum_{j=1}^{d_{l-1}} 0 \cdot x_j = 0, \qquad (7)$$

namely, $\mathbf{h}$ is zero-mean. $\qquad\square$

Therefore, the column centered constraint can ensure zero-mean property of the output, including an equivalent effect of a prior centering operation. By applying column centered constraint on a linear layer, we can form a following redundant centering.

### 3.3 REGULABLE MODULES

For more general analysis, we delve into other linear modules that only include linear transformation, for example recurrent layer with shared weights in RNN, convolution layer in CNN. We denote that the core idea of designing a constraint on any linear module is to **ensure the input weights are zero-mean**.

With the linearity, the zero-mean of input weight can always ensure the zero-mean of output, regardless of the input. In this way, we transform zero-mean property from the data to the parameter, which is always independent of the samples. In terms of the constraints and the proofs of zero-mean property for recurrent layer and convolution layer, please refer to Appendix A.3 for details.

To be mentioned, despite that self-attention module is non-linear as it has softmax operation, we can see it as a combination of linear and non-linear modules and make use of its posterior linear component—matrix multiplication of $\boldsymbol{V}$, thus construct the constraint.

Therefore, enlightened by the self-attention module, we then define *regulable modules*.

**Definition 3.** *(Regulable Module.) A regulable module is a linear module or a sub-network ended with a linear module.*

The regulable modules here include linear modules, such as linear layers, recurrent layers and convolution layers, and particular non-linear modules, such as self-attention modules. We can always find a column centered constraint for each regulable module.

**Group Normalization** We also extend the constrains and conclusion to group normalization (Wu & He, 2018)—a more general extenson of layer normalization. We demonstrated grouped column centered constraint in Appendix A.4.

Consequencely, *a regulable module under column centered constraint* can form redundant centering after it.

## 4 EQUIVALENT INFERENCE FOR PRE-TRAINED MODELS

In this section, we first propose a simple transformation to ensure the *constraint* for pre-trained models during inference, based on the analyses in Section 3. We then define foldable LN and therefore we introduce a general algorithm to detect how many LNs can be safely replaced without affecting the outcome of a model.

### 4.1 COLUMN CENTERED WEIGHT TRANSFORMATION

To achieve column centered constraint in neural network, we propose *column centered weight transformation* to ensure that the transformed weight matrix can obtain the zero-mean property of each column. Taking the linear layer for example, we have the definition as below.

**Definition 4.** *(Column Centered Weight Transformation (CCWT).) Column centered weight transformation aim to apply transformation onto weight matrix to ensure column centered constraint. We construct a specific transformation $\Psi$, change $\boldsymbol{W}$ into $\boldsymbol{W}'$, as:*

$$\boldsymbol{W}' = \Psi(\boldsymbol{W}) = \left( \boldsymbol{I} - \frac{1}{m}\mathbf{1}_m^\top \mathbf{1}_m \right) \boldsymbol{W} \tag{8}$$

*where $m$ is the output neuron number.*

Apparently, CCWT always ensures that the transformed matrix $\boldsymbol{W}'$ is under column centered constraint and form redundant centering. It is worth noting that for different regulable modules, the transformation $\Psi$ may take different forms, but the essence of its construction based on column centered constrain will not change. *We demonstrate corresponding CCWT of column centered constraints in Appendix A.3.*

## 4.2 REPLACEMENT WITH A EQUIVALENT FUNCTION

Before using the previously described method–forming redundant centering by transformation and replace LN with RMSNorm–for applications, we have to ensure that it does not apply any other effect. Here, we discuss the relationship between CCWT and centering operation in LN.

**Proposition 2.** *CCWT has and only has the same effect as centering operation in forward propagate.*

*Proof.* Here, we take a linear layer and a following LN as an example. To prove the proposition, we compute the input of scaling operation in two different models. We define model $A$ with ordinary linear layer before normal LN, model $B$ under column centered weight transformation.

In model $A$, by definition of centering operation and linear layer, we have linear layer with:

$$\mathbf{h}_A = \boldsymbol{W}_A \mathbf{x}_A, \tag{9}$$

and centering operation with:

$$\widetilde{\mathbf{h}}_A = \left( \boldsymbol{I} - \frac{1}{m}\mathbf{1}_m \mathbf{1}_m^\top \right) \mathbf{h}_A. \tag{10}$$

When in model $B$, according to the definition of column centered weight transformation $\Psi$, we have weight matrix used for calculation as:

$$\boldsymbol{W}'_B = \left( \boldsymbol{I} - \frac{1}{m}\mathbf{1}_m \mathbf{1}_m^\top \right) \boldsymbol{W}_B, \tag{11}$$

and linear layer

$$\widetilde{\mathbf{h}}_B = \mathbf{h}_B = \boldsymbol{W}'_B \mathbf{x}_B. \tag{12}$$

It is easy to identify the two forward process are the same: $\widetilde{\mathbf{h}} = \left( \boldsymbol{I} - \frac{1}{m}\mathbf{1}_m\mathbf{1}_m^\top \right) \boldsymbol{W}\mathbf{x}$. Thus we conclude that the forward process are the same. $\square$

Therefore, the CCWT has the exact functionality of centering operation in subsequent LN, obtaining redundant centering. We can thus apply CCWT onto the module and safely replace the LN with RMSNorm for inference.

To be noted, the transformation only need to be done once at the very beginning of validation, since the weight matrix will not update. Accordingly, once applied with our transformation, LN can be changed into RMSNorm without any other change in the model reducing both memory and calculation usage. Theoretically,

## 4.3 FOLDABLE LAYER NORMALIZATION IN INFERENCE

As we summarize the method of forming redundant centering and propose removing LN with RMSNorm to reduce usage, we define this simplification method as folding layer normalization, the layer normalization satisfying the requirement as *foldable layer normalization*.

**Definition 5.** *(Foldable Layer Normalization.) Given a layer normalization and its input $f(\mathbf{x}; \boldsymbol{\theta})$ from a corresponding module. We call this layer normalization foldable, if there is some map $\psi : \mathbb{R}^m \to \Theta$, subjected to*

$$RMSNorm(f(\mathbf{x}; \psi(\boldsymbol{\theta}))) = LN(f(\mathbf{x}; \boldsymbol{\theta})). \tag{13}$$

Specifically, a layer normalization is foldable, if we can use the aforementioned method to map the weight of all corresponding models into a column centered manifold space. Therefore, to fold any given LN, we would like to find out which module to be applied with the constraint.

Based on the characteristics of forward propagation, for each LN, we only need to consider the module before it. Simply, if LN only connects to one regulable module, we can apply column centered constraint. For more complex situations, such as multiple modules connected to a single LN, we can separately treat these modules, according to the distributive property of addition as demonstrate in Appendix A.3.4.

Such that, if all of these modules are regulable modules, the following LN can be folded. Based on this idea, we define the *corresponding module* for a LN to form the redundant centering.

**Definition 6.** *(Corresponding Module.) Given a neural network with layer normalizations. For the layer normalization and all the channels directly linked to it, we define all the adjacent modules as corresponding modules of this layer normalization.*

Here, we assume that all the corresponding modules only *connect* to LN. *For the commonly used models nowadays, such as transformers, all meet this requirement.* Therefore, if all the corresponding modules of a LN are regulable module and applied with CCWT, the LN will be foldable. *We notice an acceleration with a foldable LN theoretically. We include the calculation in Appendix A.8.1.*

---

**Algorithm 1** Detect foldable LayerNorm modules.

---

1: **Input:** Model $\mathcal{M}$ with input tensor $T_0^{\text{in}}$
2: **Output:** Set $\mathcal{S}$ of foldable LayerNorm modules

3: $T_0^{\text{in}}.centered \leftarrow \texttt{False}$        ▷ Set initial tensor state
4: $\mathcal{S} \leftarrow \emptyset$        ▷ Initialize set of foldable LayerNorms

5: **for** each step $t$, module $M_t \in \mathcal{M}$ **do**    ▷ Iterate through each module that tensors pass through
6:      **if** $T_t^{\text{in}} \leftarrow \sum_{i=1}^n T_i$ **then**      ▷ Residual connection where multiple tensors are combined
7:          $T_t^{\text{in}}.centered \leftarrow \bigwedge_{i=1}^n T_i.centered$      ▷ New state is the logical AND of the addends
8:      **end if**
9:      **if** $M_t = \text{LayerNorm} \land T_t^{\text{in}}.centered = \texttt{True}$ **then**
10:          $\mathcal{S} \leftarrow \mathcal{S} \cup \{M_t\}$        ▷ Mark this LayerNorm as foldable
11:      **end if**

12:      $T_t^{\text{out}} \leftarrow M_t(T_t^{\text{in}})$        ▷ The output tensor is computed by applying the module
13:      **if** $M_t \in \text{Regulable Modules}$ **then**
14:          $T_t^{\text{out}}.centered \leftarrow \texttt{True}$
15:      **else if** $M_t = \text{Dropout}$ **then**
16:          $T_t^{\text{out}}.centered \leftarrow T_t^{\text{in}}.centered$        ▷ Keep previous centered state
17:      **else**
18:          $T_t^{\text{out}}.centered \leftarrow \texttt{False}$
19:      **end if**
20: **end for**

21: **Return:** The list of foldable LayerNorm modules

---

### 4.4 ALGORITHM IN DETECTING FOLDABLE LN

When looking across the entire network, we hope to simplify some of, and even all of, the LNs in the model. Therefore, we propose an algorithm (Algorithm 1) to detect how many LNs are foldable.

To be noted, due to the residual structure and the none-zero-mean output of embedding layer, we cannot find any foldable LN in pre-norm transformer by this method. For most models that include pre-norm transformation, they often have a layer normalization after the last transformation block. Accordingly, if we add an extra centering after the embedding operation, all of the LNs will become foldable. We prove this in Appendix A.5.

We have tested the above algorithm on GPT-2 (Radford et al., 2019), BERT (Devlin et al., 2019), ViT (Dosovitskiy et al., 2021), Phi (Gunasekar et al., 2023), T5 (Raffel et al., 2020), and BLOOM (Scao et al., 2022), and found that all LN modules in these models are foldable. *We list more details in Appendix A.9*.

Moreover, in order to verify the effectiveness of this method in practical applications, we *apply our method on GPT-2, BERT and Bloom*. The results show that the replacement GPT-2 model is 10.31% faster in *total inference time (from 0.0152s to 0.0136s), and the 3 models all enjoys an acceleration of 10% to 20% in efficiency in CUDA time. For more details, please refer to Appendix A.8.2.*

## 5 TRAINING FROM SCRATCH

In this section, we focus on training the network with RMSNorm while maintaining the theoretical advantages of LN in improving the conditioning of optimization Ba et al. (2016). While this advantage of LN is not reached to a consensus in the community, we *do* observe that the centering of LN helps to stabilize the range of the output for a network in our experiments shown in the subsequent experiment.

### 5.1 OBSERVATION OF CENTERING

Here, we conduct ablation experiments to show how centering helps control the range of the output for a network. We train MLPs of different depths using both LN and RMSNorm, under the classification task of CIFAR-10 and MNIST, as detailed in Appendix A.10.1. We monitor the parameters and input of each layer in each epoch. The results are shown in Figure 1. We find that the norm of input of the last layer is better controlled in a smaller range by LN during the whole training process. Moreover, the change of input mean and norm are more intense under RMSNorm without centering operation. Our experiment suggests that the centering helps to stabilize the range of the output for a network.

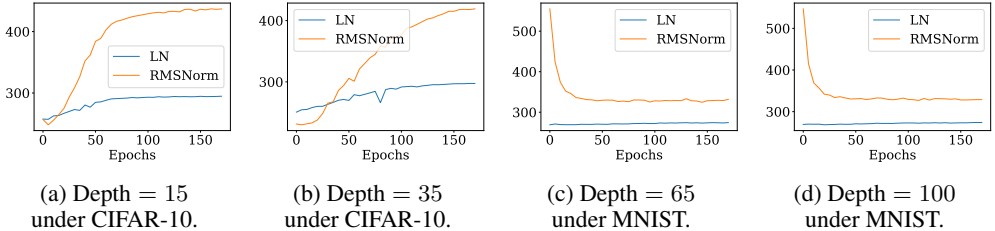

| (a) Depth = 15 under CIFAR-10. | (b) Depth = 35 under CIFAR-10. | (c) Depth = 65 under MNIST. | (d) Depth = 100 under MNIST. |

Figure 1: Norm of input of last layer in different depth of MLP.

### 5.2 COLUMN BASED WEIGHT CENTERING

Based on the previous analyses, the key to remove the centering of LN during training is to ensure that all the regulable modules of the corresponding modules of LN is under the column centered constraints during the course of training.

To achieve this idea, we introduce re-parameterization. Re-parameterization is a transformation of a model's parameter space Nowlan et al. (1998), aimed to maintain a certain property of the parameters in the training process. We use re-parameterization to ensure that the weight matrix for calculation can obtain column centered constraint throughout training. We thus propose *column based weight centering* based on the re-parameterization.

**Definition 7.** *(Column Based Weight Centering (CBWC).) Column based weight centering is a re-parameterization, applying a proxy parameter $\boldsymbol{W}$ to control the transformed matrix $\boldsymbol{V}$. We construct a specific transformation $\Psi$*

$$\boldsymbol{V} = \Psi(\boldsymbol{W}) = \left(\boldsymbol{I} - \frac{1}{m}\mathbf{1}_m^\top\mathbf{1}_m\right)\boldsymbol{W} \tag{14}$$

*where $m$ is the output neuron number.*

*In back propagation, we update gradient of the proxy parameter $\boldsymbol{W}$ from the transformed matrix $\boldsymbol{V}$ back through the transformation. We have backward transformation $\boldsymbol{\psi}$*

$$\frac{\partial \mathcal{L}}{\partial \boldsymbol{W}} = \boldsymbol{\psi}\left(\frac{\partial \mathcal{L}}{\partial \boldsymbol{V}}\right) = \left(\boldsymbol{I} - \frac{1}{m}\mathbf{1}_m^\top\mathbf{1}_m\right)^\top \frac{\partial \mathcal{L}}{\partial \boldsymbol{V}}. \tag{15}$$

Apparently, CBWC always ensures that the transformed matrix $\boldsymbol{W}$ is under column centered constraint. Therefore, we can apply CBWC to obtain redundant centering during training, *as a necessary precondition of folding LN. Here, we prove the centering of a foldable LN can be removed by introducing CBWC*, without affecting the training dynamics.

**Proposition 3.** *(Equivalent Optimization Process.) The optimization processes of a foldable LN and a RMSNorm combining CBWC are identical.*

*Proof.* It is easy to identify the forward processes of a foldable LN and a RMSNorm combining CBWC are the same, as we discussed in Proposition 2. Similarly, since both have the same learnable parameters, outputs and gradients of parameters, the back propagate processes are also the same. For more details, please refer to Appendix A.7. □

Similar to the analyses for inference in Section 4, we can further consider the equivalence of optimization process between a network with foldable LN and a RMSNorm combining CBWC. We provide the analysis for the Transformer in Appendix A.6. *Therefore, we introduce 'CBWC+RMSNorm' as a alterative training method with origin LN.* These two sets of parameters can be converted into each other, due to their equivalent optimization process*, more details are in Section 5.4*.

**Computational Complexity Analysis**  One advantage of '*CBWC+RMSNorm*' over LN is the computational efficiency. In the widely used Transformer model, with a batch size denoted by $b$ and sequence length denoted by $s$, the dimension of a word is represented by $d$, and the weight matrix is denoted as $\mathbf{W} \in \mathbb{R}^{d \times p}$. Considering one epoch training with $B$ samples, centering over the samples (the centering of LN) has a computational cost of approximately $O(Bsd)$ while *CBWC* incurs a computational cost of $O(Bdp/b)$. We can find that *CBWC* is more efficient, if $s * b > p$. In practical situation, $s * b$ is much larger than $p$, especially in the scenario with long context learning.

## 5.3 EMPIRICAL STUDY

Even though we provide a theoretical equivalence for optimization between a foldable LN and '*CBWC+RMSNorm*' during training, the model architecture is likely to have a bunch of Dropout layers (Vaswani et al., 2017) between linear modules and layer normalization. The training mode of Dropout will disrupting the zero-mean property[3] This results in that the centering operation and CBWC is not theoretically equivalent while training.

Here, we conduct experiments to empirically validate the effectiveness of our proposed 'RMSNorm +CBWC' in Transformer for text translation, text classification tasks *and image classification*.

**Text Translation**  In this part, we apply CBWC to the transformer architecture for text translation task. We investigate both the training and inference performances on Multi30K (Elliott et al., 2016) dataset. We follow the same experimental protocol as (Vaswani et al., 2017) and apply CBWC and replace RMSNorm with LN. Here, we use training loss to measure performances of training and bilingual evaluation understudy (BLEU) (Kishore Papineni & Zhu, 2002) scores in inference.

We compare three models: the baseline transformer model with LN, the variant with RMSNorm as well as the variant applying both CBWC and RMSNorm. All models are trained for 1000 epochs. From Figure 2, after the first 100 epochs, the baseline model and our method still maintain a close alignment, while both notably outperform the model applying only RMSNorm. Notably, our method exhibits slightly inferior performance in terms of both training loss and BLEU scores compared to the baseline model. As mentioned earlier, we attribute this phenomenon to the abundant presence of dropout layers in transformer architecture.

---

[3]The inference mode of Dropout can be viewed as a scalar, which doe not affect the zero-mean property.

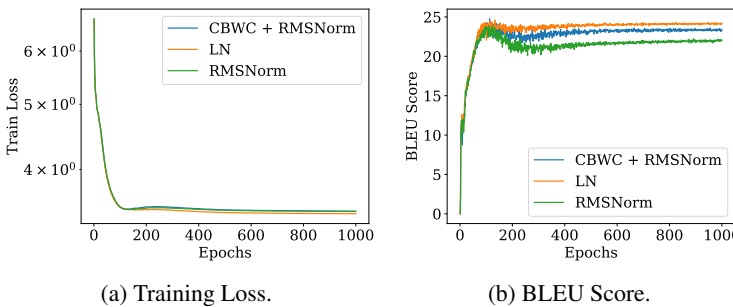

(a) Training Loss.
(b) BLEU Score.

Figure 2: Results of transformer models for text translation task.

**Text Classification** For experiments on text classification tasks based on transformer architecture, we selected the AG News (Zhang et al., 2015) dataset for our experiments. The experiment settings are the same as in the text translation task. We use loss and accuracy to measure performances of training and inference.

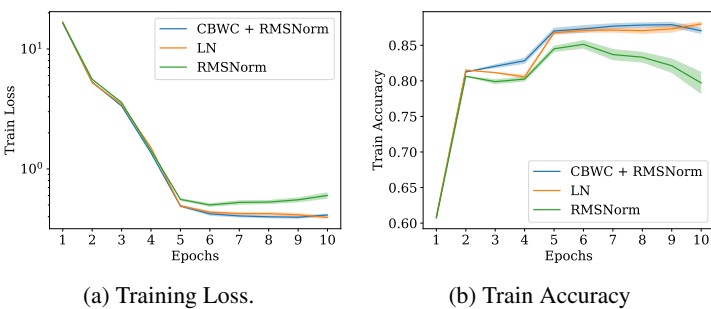

(a) Training Loss.
(b) Train Accuracy

Figure 3: Results of transformer models for text classification task when training with standard deviation shown in shaded region.

All models are trained for 10 epochs. From Figure 3, the baseline model and the our method outperform the model applying only RMSNorm when training. Additionally, the model applying only RMSNorm has an unstable inference performance.

We also compare features among transformer models when applying LN, only RMSNorm and both CBWC and RMSNorm. We show test loss and test accuracy with standard deviation. The results are averaged by five random seeds respectively and shown in Table 1 The test loss and test accuracy of our method are close to the baseline model applying LN, outperforming the model applying only RMSNorm.

Table 1: Results of transformer models for text classification task when inference.

| Model | Test Loss | Test Acc |
|---|---|---|
| LN | $\mathbf{0.4721} \pm 0.0100$ | $\mathbf{85.21\%} \pm 0.35\%$ |
| RMSNorm | $\mathbf{0.6630} \pm 0.0357$ | $\mathbf{76.57\%} \pm 1.77\%$ |
| *CBWC+RMSNorm* | $\mathbf{0.4728} \pm 0.0066$ | $\mathbf{85.12\%} \pm 0.23\%$ |

*Image Classification We conduct image classification tasks based on SWIN (Liu et al., 2021) on Imagenet100 (Chun-Hsiao Yeh, 2022). Here we apply CBWC and an addition centering operation after embedding layer, and replace LN with RMSNorm. We measure the performances with accuracy for train and test, and evaluate the efficiency by measuring forward pass time, backward propagate time and validation time. For more details, please refer to Appendix A.10.2.*

*All models are trained for 40 epochs and averaged by 4 random seeds. Our method have a improvement in time usage while it performance outperforming both RMS and LN in test.*

*Here, we conduct the experiments on a rather small patch size, which leads to a long sequence. Such that our method advantage in training stage. To be noted, despite our method has a reduction in*

Table 2: *Time and performance results of SWIN on Imagenet100.*

| Model | LN | CBWC+RMS | Acceleration | RMS-only | Acceleration |
|---|---|---|---|---|---|
| Train-FP (1e-6 s) | 80991.77 | 78065.93 | 3.61% | 77209.91 | 4.67% |
| Train-BP (1e-6 s) | 37055.36 | 33737.88 | 8.95% | 30429.37 | 17.88% |
| Eval (1e-6 s) | 310678.69 | 298863.31 | 3.80% | 296510.13 | 4.56% |
| Test ACC1 (%) | 90.5429 | 90.5853 | 0.05% | 90.5636 | 0.02% |
| Test ACC5 (%) | 95.7495 | 95.8777 | 0.13% | 95.7889 | 0.04% |

*training accuracy, it has a more stable training process, more details are in Appendix A.10.2. To be noted, our method and RMSNorm differ in validation time usage despite theoretical equivalence. We attribute this phenomenon to the extra CCWT at the very beginning of each validation process.*

## 5.4 CONTINUE LEARNING

In practical applications, we extensively use pre-trained models for both inference and continue learning. Due to the same optimization process of LN and '*CBWC+RMSNorm*', theoretically, a pre-trained model can continue to train with '*CBWC+RMSNorm*' by replacing the LN. Here, We conduct experiments to verify it. We place the proxy parameter in CBWC with the weight matrix of the pre-trained model and use RMSNorm in place of LN. Under the same training settings described in Appendix A.10.3, the weight matrices of the two models, with and without our method, theoretically undergo the same learning process.

We find out that the proxy parameter $W_A$ under our method and the origin weight matrix $W_B$ almost identical — the differences are smaller than $10^{-5}$, which can be seen as a calculation error — after 40 epochs of training under the same random seed. We draw the conclusion that model $A$ and model $B$ have the same optimization process.

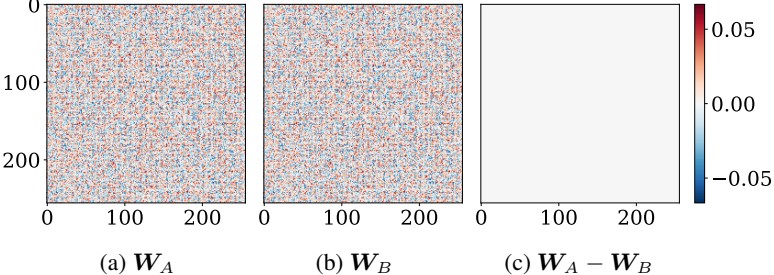

(a) $W_A$      (b) $W_B$      (c) $W_A - W_B$

Figure 4: Comparison of $W_A$ and $W_B$.

## 6 CONCLUSION

This paper provided the framework by rigorous definition and derivation, under which a DNN with LN can be equivalently transfer to a network with RMSNorm. We showed how the centering operation of LN can be removed both in inference and training, by introducing the proposed column centered weight transformation (CCWT) and column based weight centering (CBWC). The proposed CCWT can be directly applied to various pre-trained large language models (LLMs) and large vision language models (VLMs) with LN, enabling an immediate reduction in computation cost but with an equivalent forward pass during inference. We hope our method can benefit the LLMs and VLMs community.

**Limitation and Future Work** In practical application scenarios, there are a large number of LLMs and VLMs with LN, but the number of large models we have analyzed in this paper is relatively small. In future work, we will develop general tools to detect foldable LNs in pre-trained models and replace these LNs with RMSNorm automatically. Morever, in real neural networks, there are some modules that our method cannot implement, such as dropout layers. *This may affect the effectiveness and utility of our simplification method, which is determined by the construction of the model. In future work, we expect to have more detailed analysis of every models on how much does this method improve the effect.*

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

# A APPENDIX / SUPPLEMENTAL MATERIAL

## A.1 SKETCH MAP

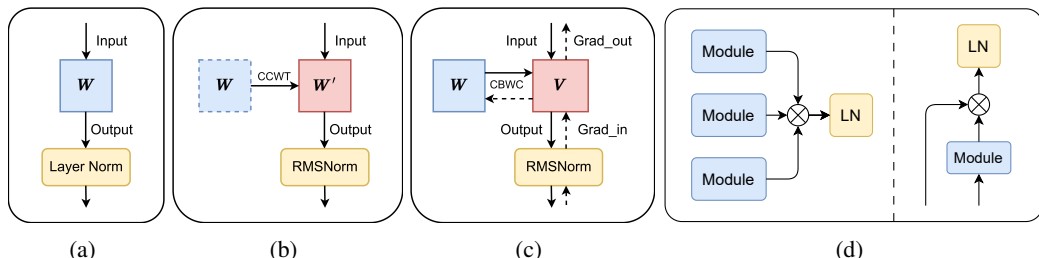

$$(a) \qquad\qquad (b) \qquad\qquad (c) \qquad\qquad (d)$$

Figure 5: Sketch maps of *the concepts* in this paper. (a) Origin Model. (b) *CCWT*. (c) *CBWC*. (d) Corresponding Modules.

## A.2 BIAS IN THE WEIGHT MATRIX

We consider to remove the bias $\mathbf{b} = [b_1, b_2, \ldots, b_m]^\top \in \mathbb{R}^{m \times 1}$ in $\mathbf{h} = \boldsymbol{W}\mathbf{x} + \mathbf{b}$. We add an additional dimension to $\mathbf{x}$, which turns it into $\mathbf{x}' = [x_1, x_2, \ldots, x_d, 1]^\top \in \mathbb{R}^{(d+1) \times 1}$, and an additional column in $\boldsymbol{W}' \in \mathbb{R}^{m \times (d+1)}$, where

$$\boldsymbol{W}' = \begin{bmatrix} w_{1,1} & w_{1,2} & \cdots & w_{1,d} & b_1 \\ w_{2,1} & w_{2,2} & \cdots & w_{2,d} & b_2 \\ \vdots & \vdots & \ddots & \vdots & \vdots \\ w_{n,1} & w_{m,2} & \cdots & w_{m,d} & b_m \end{bmatrix}. \tag{16}$$

We can add the bias into the weight in this way.

## A.3 THE PROOF OF REDUNDANT CENTERING

In this section, we demonstrate the column centered constraint of the modules, and give the proof that they ensure the output of module centralized before layer normalization.

### A.3.1 RECURRENT NEURAL NETWORK

Despite linearity, the recurrent neural network is different from origin linear layer with its recurrent connection and shared weight matrix. Due to the fact that our constraints are independent of input and output, the parameter sharing may be excluded from our consideration. As for the recurrent connection, the weights for ordinary input and recurrent input can be seen as two linear layer.

For the $l$-th layer and $t$-th time step of the network, we define the input as $\mathbf{x}_t^{l-1} \in \mathbb{R}^{d_{l-1}}$, the recurrent input as $\mathbf{h}_{t-1}^l \in \mathbb{R}^{d_l}$ and output of hidden layer as $\mathbf{c}_t^l$. We have weight matrix $\boldsymbol{W}_v \in \mathbb{R}^{d_{l-1} \times d_l}$ and $\boldsymbol{W}_h \in \mathbb{R}^{d_l \times d_l}$, which is shared among all time steps. We define $\boldsymbol{W} = [\boldsymbol{W}_v, \boldsymbol{W}_h]$. We have the constraint:

$$\boldsymbol{W}_0 \in \Gamma_{rnn} = \left\{ \boldsymbol{W} : \sum_{i=1}^{d_l} w_{i,j}^v = 0, \sum_{i=1}^{d_l} w_{i,k}^h = 0, \ j = 1, 2, \ldots, d_{l-1}, \ k = 1, 2, \ldots, d_l \right\}. \tag{17}$$

In this recurrent neural network, we have output of hidden layer as $\mathbf{c}_t = \boldsymbol{W}_v \mathbf{x}_t^{l-1} + \boldsymbol{W}_h \mathbf{h}_{t-1}^l$. Under the constrain of Eqn. 17, we have the mean of output with

$$
\begin{aligned}
\mu_t^c &= \frac{1}{d_l} \sum_{i=1}^{d_l} \left( \sum_{j=1}^{d_{l-1}} w_{i,j}^v x_j + \sum_{k=1}^{d_l} w_{i,k}^h h_k \right) \\
&= \frac{1}{d_l} \left( \sum_{j=1}^{d_{l-1}} \left( \sum_{i=1}^{d_l} w_{i,j}^v \right) x_j + \sum_{k=1}^{d_l} \left( \sum_{i=1}^{d_l} w_{i,k}^h \right) h_k \right) \\
&= \frac{1}{d_l} \left( \sum_{j=1}^{d_{l-1}} 0 \cdot x_j + \sum_{k=1}^{d_l} 0 \cdot h_k \right) = 0.
\end{aligned}
\tag{18}
$$

Thus, for shared weight matrix for both input from last layer and from last time step, applying constraint on them centralize the output of the hidden layer.

*Accordingly, we have the transformation $\Phi_{rnn,v}, \Phi_{rnn,h}$ of the CCWT on recurrent neural Network, as follow:*

$$
\begin{aligned}
W^v &= \Phi_{rnn,v}(W^v) = (I - \frac{1}{m_v} \mathbf{1}_{m_v}^\top \mathbf{1}_{m_v}) W^v \\
W^h &= \Phi_{rnn,h}(W^h) = (I - \frac{1}{m_h} \mathbf{1}_{m_h}^\top \mathbf{1}_{m_h}) W^h
\end{aligned}
\tag{19}
$$

### A.3.2 CONVOLUTION LAYER

Under the circumstances of the convolution layer, the convolutional kernel can be regarded as a combination of a set of shared weights. All of them should fulfill the constraint of linear layer. We hence use vector to denote the elements among different channel among the kernel.

We denote the input tensor $\mathbf{x} \in \mathbb{R}^{d_{l-1} \times h \times w}$ and the output tensor $\boldsymbol{H} \in \mathbb{R}^{d_l \times h' \times w'}$. We have convolution kernels $\boldsymbol{W} \in \mathbb{R}^{d_l \times d_{l-1} \times F_h \times F_w}$. We have the constraint:

$$
\boldsymbol{W}_0 \in \Gamma_{cnn} = \left\{ \boldsymbol{W} : \sum_{i=1}^{d_l} \mathbf{w}_{i,j} = 0, \ j = 1, 2, \ldots, d_{l-1} \right\}.
\tag{20}
$$

For every channel of output tensor $\boldsymbol{H}_i \in \mathbb{R}^{h' \times w'}$ $(i = 1, \ldots, d_l)$ and corresponding convolution kernel $\mathbf{w}_{i,j} \in \mathbb{R}^{F_h \times F_w}$ $(i = 1, \ldots, d_l, \ j = 1, \ldots, d_{l-1})$, we have

$$
\boldsymbol{H}_i = \sum_{j=1}^{d_{l-1}} \mathbf{x}_j * \mathbf{w}_{i,j}.
\tag{21}
$$

Due to convolution operation, we have $a * (b + c) = a * b + a * c$. Thus, under the constrain of Eqn.20 we have:

$$
\begin{aligned}
\mu_h &= \frac{1}{d_l} \sum_{i=1}^{d_l} \boldsymbol{H}_i = \frac{1}{d_l} \sum_{i=1}^{d_l} \sum_{j=1}^{d_{l-1}} \mathbf{x}_j * \mathbf{w}_{i,j} = \frac{1}{d_l} \sum_{j=1}^{d_{l-1}} \mathbf{x}_j * \left( \sum_{i=1}^{d_l} \mathbf{w}_{i,k} \right) \\
&= \frac{1}{d_l} \sum_{j=1}^{d_{l-1}} \mathbf{x}_j * 0 = 0.
\end{aligned}
\tag{22}
$$

It thus can be seen that the column centered constraint on the convolution kernels achieves the effect of the centering of the layer normalization.

*Accordingly, we have the transformation $\Phi_{cnn}$ of the CCWT on convolution layers, as follow:*

$$W = \Phi_{cnn}(W) = (I - \frac{1}{h \times w}\mathbf{1}_{h \times w}^{\top}\mathbf{1}_{h \times w})W \tag{23}$$

*To be noted, the tensor $W$ here is a four-dimension tensor. The transformation here is to do centering on its second dimension.*

### A.3.3 ATTENTION

To be mentioned, despite that self-attention module is non-linear as it has softmax operation, we make use of its posterior linear component thus construct the constraint on self-attention module.

Self-attention can extract similar structure with linear layers, but it is more complicated. For a sampled input $\mathbf{x} \in \mathbb{R}^{n \times d}$, we apply three different learnable weight matrices $\boldsymbol{Q}, \boldsymbol{K} \in \mathbb{R}^{d \times d_k}$, $\boldsymbol{V} \in \mathbb{R}^{d \times d_v}$ and have three input matrices $\boldsymbol{H}_Q, \boldsymbol{H}_K \in \mathbb{R}^{n \times d_k}$, $\boldsymbol{H}_V \in \mathbb{R}^{n \times d_v}$ with

$$\boldsymbol{H}_Q = \mathbf{x} \cdot \boldsymbol{Q}, \quad \boldsymbol{H}_K = \mathbf{x} \cdot \boldsymbol{K}, \quad \boldsymbol{H}_V = \mathbf{x} \cdot \boldsymbol{V}. \tag{24}$$

According to scaled dot-product attention, we have:

$$\text{Attention}(\boldsymbol{H}_Q, \boldsymbol{H}_K, \boldsymbol{H}_V) = \text{softmax}\left(\frac{\boldsymbol{H}_Q \boldsymbol{H}_K^{\top}}{\sqrt{d_k}}\right)\boldsymbol{H}_V. \tag{25}$$

The linear layer of self attention change the left multiplication of the weight matrix into right multiplication. Thus our constraint changes into row centered constraint, with equation

$$\boldsymbol{W}_0 \in \Gamma_{trans} = \left\{ \boldsymbol{W} : \sum_{k=1}^{d_v} v_{j,k} = 0, \ i = 1, 2, \ldots, d_{l-1} \right\}. \tag{26}$$

We define $\boldsymbol{B} = \text{softmax}\left(\frac{\boldsymbol{H}_Q \boldsymbol{H}_K^{\top}}{\sqrt{d_k}}\right) \in \mathbb{R}^{n \times n}$. Since $\boldsymbol{H}_V = \mathbf{x} \cdot \boldsymbol{V}$, we have

$$\boldsymbol{H}_V^{(i,j)} = \sum_{k=1}^{d} x_{i,k} \cdot v_{k,j} \quad (i = 1, \ldots, n, \ j = 1, \ldots, d_v),$$
$$\text{Attention}(\boldsymbol{H}_Q, \boldsymbol{H}_K, \boldsymbol{H}_V) = \boldsymbol{B} \cdot \boldsymbol{H}_V. \tag{27}$$

By Eqn.26 and Eqn.27, we have

$$\mu_a = \sum_{b=1}^{d_v}(\boldsymbol{B}\boldsymbol{H}_V)_{(a,b)} = \sum_{b=1}^{d_v}\sum_{j=1}^{n} b_{a,j} \cdot \boldsymbol{H}_{V(j,b)} = \sum_{b=1}^{d_v}\sum_{j=1}^{n} b_{a,j}\left(\sum_{k=1}^{d} x_{j,k} \cdot v_{k,b}\right)$$
$$= \sum_{b=1}^{d_v}\sum_{j=1}^{n}\sum_{k=1}^{d} b_{a,j} \cdot x_{j,k} \cdot v_{k,b} = \sum_{j=1}^{n}\sum_{k=1}^{d} b_{a,j} \cdot x_{j,k}\left(\sum_{b=1}^{d_v} v_{k,b}\right) \tag{28}$$
$$= \sum_{j=1}^{n}\sum_{k=1}^{d} b_{a,j} \cdot x_{j,k} \cdot 0 = 0.$$

Thus, we only need to apply constraint to the weight matrix $\boldsymbol{V}$, which generated the $\boldsymbol{H}_V$. With this constraint, the output of whole scaled dot-product attention can be centralized.

*Accordingly, we have the transformation $\Phi_{trans}$ of the CCWT on self-attention modules, as follow:*

$$W^v = \Phi_{trans}(W^v) = (I - \frac{1}{m_v}\mathbf{1}_{m_v}^{\top}\mathbf{1}_{m_v})W^v \tag{29}$$

Moreover, for multi-head attention, applying constraint onto the linear on the last only can ensure zero-mean output.

Notice that the conclusion only fit post-LN transformer.

### A.3.4 RESIDUAL STRUCTURE

For a residual structure, we define the input $\mathbf{x}$ and the output $\mathbf{y}$, as shown below

$$\mathbf{y} = \mathcal{F}(\mathbf{x}) + \mathcal{G}(\mathbf{x}), \tag{30}$$

where we have $\mathcal{F}$ as a consistent of two linear layer and one ReLU function and $\mathcal{G}(\mathbf{x}) = \mathbf{x}$ in origin residual structure.

Due to the complexity of $\mathcal{F}(\cdot)$, it is intuitively difficult to construct a constraint on this function to eliminate the mean of $\mathcal{G}(\mathbf{x})$. We treat the two terms separately, and apply constraint based on their content.

In the origin residual structure, for $\mathcal{G}(\mathbf{x}) = \mathbf{x}$, if it is already centralized, then we do not need to apply any constraint. If not, we can see it as $\boldsymbol{I}\mathbf{x}$, thus apply linear layer constraint on $\boldsymbol{I}$. To be specific, change $\mathbf{x}$ into $(\boldsymbol{I} - \frac{1}{m}\mathbf{1}_m\mathbf{1}_m^\top)\mathbf{x}$.

### A.4 GROUPED COLUMN CENTERED CONSTRAINT FOR GN

We extend the conclusion to Group Normalization (GN). (Wu & He, 2018)

Group Normalization is first defined on channel dimension for convolutional input $\boldsymbol{X} \in \mathbb{R}^{d \times h \times w}$. So the Group Normalization here is more similar to grouped Layer Normalization, with the definition below:

**Definition 8.** *(Group Normalization (GN).) Suppose the number of groups is $g$, and $d = g \times c$. Let $\mathbf{x} = [\boldsymbol{z}_1^\top, \ldots, \boldsymbol{z}_g^\top]^\top$, where $\boldsymbol{z}_i = [z_{i1}, \ldots, z_{ic}]^\top, (i = 1, \ldots, g)$. Assume $\mathbf{x} = [x_1, \ldots, x_d]^\top$, we denote that $z_{ij} = x_{(i-1) \times c + j}$. Let $\hat{\mathbf{x}} = GN(\mathbf{x})$, where $GN(\cdot)$ denotes the Group Normalization operation. GN can be calculated by $\mu_i = (z_{i1} + \cdots + z_{ic})/c$, $\sigma_i^2 = [(z_{i1} - \mu_i)^2 + \cdots + (z_{ic} - \mu_i)^2]/c$, and then $\hat{z}_{ij} = (z_{ij} - \mu_i)/\sigma_i$. Thus, we have $\hat{\mathbf{x}} = [\hat{\boldsymbol{z}}_1^\top, \ldots, \hat{\boldsymbol{z}}_g^\top]^\top$, where $\hat{\boldsymbol{z}}_i = LN(\boldsymbol{z}_i), (i = 1, \ldots, g)$.*

*For every input, we divide the neurons into groups and apply normalization in every group. Thus the centering step in this normalization is to ensure the output sum of all neurons in each group is zero. For a sampled input $\mathbf{h} = [h_1, h_2, \ldots, h_m]^\top$, for $g$ groups and $c$ channels in every group ($g \times c = m$), we have*

$$\mu_{hj} = \frac{1}{c} \sum_{i=1}^{c} h_{ji} = 0 \quad (j = 1, \ldots, g). \tag{31}$$

So for MLP, we have column centered constraint for GN:

$$\boldsymbol{W}_0 \in \Sigma_{GN} = \left\{ \boldsymbol{W} : \sum_{k=1}^{c} w_{j,(k+c \times i)} = 0, \ i = 1, 2, \ldots, g, \ j = 1, 2, \ldots, d \right\}. \tag{32}$$

Given $\mathbf{h} = \mathbf{W}\mathbf{x}$, for the $i$-th neuron output $h_i$ in $j$-th group of $\mathbf{h}$, we have

$$h_i = \sum_{k=1}^{d} w_{i,j} \cdot x_j. \tag{33}$$

Under the constrain of Eqn.32, we have:

$$\mu_{hj} = \frac{1}{c} \sum_{i=1}^{c} h_{ji} = \frac{1}{c} \sum_{i=1}^{c} \sum_{j=1}^{d} w_{i,j} \cdot x_j = \frac{1}{c} \sum_{j=1}^{d} \left( \sum_{i=1}^{c} w_{i,j} \right) x_j = \frac{1}{c} \sum_{j=1}^{d} 0 \cdot x_j = 0. \tag{34}$$

Thus, we replace centering step of GN with grouped column centering constraint.

To be mentioned, the core idea of designing a constraint is to ensure every group of input weight is zero-mean.

*For the transformation $\Phi_{GN}$ of the CCWT on a normal linear layer under GroupNorm, we have:*

$$W = \Phi_{GN}(W) = (I - A)W. \tag{35}$$

*A is a matrix that we construct with the equation below:*

$$A = I - \frac{1}{c} \sum_{i=0}^{d-1} \mathbf{1}_{(c,i\times c)}^{\top} \mathbf{1}_{(c,i\times c)},\tag{36}$$

*where $\mathbf{1}_{(c,i\times c)}$ refer to a vector whose elements are all zero except that the $(i \times c)$-th element to $(i \times c + c)$-th element are ones. Specifically, $A$ is a matrix with its diagonal arrayed with $c \times c$ matrices of ones.*

### A.5 THE PROOF OF PRE-LN TRANSFORMER

As we analysis, LNs in pre-norm transformer cannot be safely replaced by RMSNorm, due to the none-zero-mean of embedding layer and residual structure.

However, for most models like GPT2, there is a LN after last transformer block. LNs can thus be replaced if we add a centering in the front of block.

For the output of GPT2 model $x_{t+1}$, we have

$$x_{t+1} = x_t + h_{t+1},\tag{37}$$

here, $h_{t+1}$ refers to the output of the branch. The input of branch is $x_t$ which is directly input into a LN.

If we place a centering before the block, replace LN with RMSNorm and add CBWC onto the branch. We have the output of GPT2 model

$$x'_{t+1} = F_c(x_t) + F_c(h_{t+1}),\tag{38}$$

where $F_c$ refers to centering fuction.

We have

$$F_c(x_{t+1}) = F_c(x_t) + F_c(h_{t+1}) = x'_{t+1}.\tag{39}$$

Additionally, the input of scaling of LN on the branch are both $F_c(x_t)$.

Thus, we prove that the method can make all of the LN in GPT2 foldable.

### A.6 POST-LN TRANSFORMER BLOCK IN CONTINUE TRAINING OF PRE-TRAINED MODEL

*Proof.* To proof the proposition, we compute the input of scaling operation in layer normalizations. Similarly, we only need to focus on the forward process, since the equivalent forward process can lead to equivalent backward process.

For the origin post-LN transformer (without dropout which is applied in practice), where layer norm is located after self-attention and positional-wise fully connected feed-forward network.

To be denoted, the samples in transformer are row vectors instead of column vectors as we defined in Section 2. Moreover, we **assume** the input $\mathbf{x}$ is centralized as $\mathbf{x}(\frac{1}{m}\mathbf{1}_m\mathbf{1}_m^{\top}) = 0$ as it is often connected from the output of an LN.

Firstly, for a self-attention, we have the input of LN with the equation:

$$\widetilde{\mathbf{x}} = \mathbf{x} + \text{softmax}\left(\frac{\boldsymbol{Q}\boldsymbol{K}^{\top}}{\sqrt{d_k}}\right)\boldsymbol{V}.\tag{40}$$

We denote $\boldsymbol{Q}, \boldsymbol{K}, \boldsymbol{V}$ are generated by $\boldsymbol{W}^Q, \boldsymbol{W}^K, \boldsymbol{W}^V$, with equation $\boldsymbol{M} = \mathbf{x}\boldsymbol{W}^M, \boldsymbol{M} \in \{Q, K, V\}$. To simplify expression, we set $\boldsymbol{B} = \text{softmax}\left(\frac{\boldsymbol{Q}\boldsymbol{K}^{\top}}{\sqrt{d_k}}\right)$, thus we have

$$\mathbf{h} = \mathbf{x} + \boldsymbol{B}\mathbf{x}\boldsymbol{W}^V.\tag{41}$$

In a self-attention module with ordinary linear module before normal LN, by definition we have

$$\begin{cases} \mathbf{h} = \mathbf{x} + \boldsymbol{B}\mathbf{x}\boldsymbol{W}^V \\ \widetilde{\mathbf{h}} = \mathbf{h}\left(\boldsymbol{I} - \frac{1}{m}\mathbf{1}_m\mathbf{1}_m^{\top}\right). \end{cases}\tag{42}$$

When in a self-attention module with CBWC before RMSNorm, by definition we have

$$
\begin{cases}
\boldsymbol{V}^V = \boldsymbol{W}^V \left(\boldsymbol{I} - \frac{1}{m}\mathbf{1}_m\mathbf{1}_m^\top\right) \\
\widetilde{\mathbf{h}} = \mathbf{x} + \boldsymbol{B}\mathbf{x}\boldsymbol{V}^V.
\end{cases}
\tag{43}
$$

It is easy to identify the two forward process are the same:

$$
\widetilde{\mathbf{h}} = (\mathbf{x} + \boldsymbol{B}\mathbf{x})\left(\boldsymbol{I} - \frac{1}{m}\mathbf{1}_m\mathbf{1}_m^\top\right) = \mathbf{x} + \boldsymbol{B}\mathbf{x}\boldsymbol{W}^V\left(\boldsymbol{I} - \frac{1}{m}\mathbf{1}_m\mathbf{1}_m^\top\right).
\tag{44}
$$

Further, for a positional-wise fully connected feed-forward network, we have the input of LN with the equation:

$$
\widetilde{\mathbf{x}} = \mathbf{x} + \mathbf{x}\boldsymbol{W}.
\tag{45}
$$

In a positional-wise fully connected feed-forward network with ordinary linear module before normal LN, by definition we have

$$
\begin{cases}
\mathbf{h} = \mathbf{x} + \mathbf{x}\boldsymbol{W} \\
\widetilde{\mathbf{h}} = \mathbf{h}\left(\boldsymbol{I} - \frac{1}{m}\mathbf{1}_m\mathbf{1}_m^\top\right).
\end{cases}
\tag{46}
$$

When in a positional-wise fully connected feed-forward network with CBWC before RMSNorm, by definition we have

$$
\begin{cases}
\boldsymbol{V} = \boldsymbol{W}\left(\boldsymbol{I} - \frac{1}{m}\mathbf{1}_m\mathbf{1}_m^\top\right) \\
\widetilde{\mathbf{h}} = \mathbf{x} + \mathbf{x}\boldsymbol{V}.
\end{cases}
\tag{47}
$$

It is easy to identify the two forward process are the same: $\widetilde{\mathbf{h}} = (\mathbf{x} + \mathbf{x})\left(\boldsymbol{I} - \frac{1}{m}\mathbf{1}_m\mathbf{1}_m^\top\right) = \mathbf{x} + \mathbf{x}\boldsymbol{W}\left(\boldsymbol{I} - \frac{1}{m}\mathbf{1}_m\mathbf{1}_m^\top\right)$.

Since the processes also have the same parameters $\boldsymbol{W}$, the back propagate processes are also the same, in mathematics.

Thus, we conclude that the optimization process are the same. $\square$

Accordingly, we have '*CBWC+RMSNorm*' have the same effect with the origin linear layer and LN in optimization process, which means the same result, thus same gradient and same parameter updating.

### A.7 PROOF OF BACK PROPAGATE

*Proof.* To prove the proposition, we compute the gradient of scaling operation to input of linear layer in two different models. We define an MLP with ordinary linear layer before normal layer normalization as model $A$, an MLP under CBWC with RMSNorm as model $B$.

In model $A$, we have back propagate process as

$$
\frac{\partial \mathcal{L}}{\partial \mathbf{h}_A} = \left(\boldsymbol{I} - \frac{1}{m}\mathbf{1}_m\mathbf{1}_m^\top\right)^\top \frac{\partial \mathcal{L}}{\partial \widetilde{\mathbf{h}}_A}, \qquad \frac{\partial \mathcal{L}}{\partial \mathbf{x}_A} = \boldsymbol{W}_A^\top \frac{\partial \mathcal{L}}{\partial \mathbf{h}_A}, \qquad \frac{\partial \mathcal{L}}{\partial \boldsymbol{W}_A} = \frac{\partial \mathcal{L}}{\partial \mathbf{h}_A}\mathbf{x}_A^\top.
\tag{48}
$$

When in model $B$, according to the definition of backward transformation $\boldsymbol{\psi}$ in CBWC, similarly we have

$$
\frac{\partial \mathcal{L}}{\partial \mathbf{x}_B} = \boldsymbol{V}_B \frac{\partial \mathcal{L}}{\partial \widetilde{\mathbf{h}}_B}, \qquad \frac{\partial \mathcal{L}}{\partial \boldsymbol{V}_B} = \frac{\partial \mathcal{L}}{\partial \widetilde{\mathbf{h}}_B}\mathbf{x}_B^\top, \qquad \frac{\partial \mathcal{L}}{\partial \boldsymbol{W}_B} = \left(\boldsymbol{I} - \frac{1}{m}\mathbf{1}_m\mathbf{1}_m^\top\right)^\top \frac{\partial \mathcal{L}}{\partial \boldsymbol{V}_B}.
\tag{49}
$$

It is easy to identify the two back propagate process are the same:

$$
\frac{\partial \mathcal{L}}{\partial \mathbf{x}} = \left(\boldsymbol{I} - \frac{1}{m}\mathbf{1}_m\mathbf{1}_m^\top\right)^\top \boldsymbol{W}^\top \frac{\partial \mathcal{L}}{\partial \widetilde{\mathbf{h}}}, \qquad \frac{\partial \mathcal{L}}{\partial \boldsymbol{W}} = \left(\boldsymbol{I} - \frac{1}{m}\mathbf{1}_m\mathbf{1}_m^\top\right)^\top \frac{\partial \mathcal{L}}{\partial \widetilde{\mathbf{h}}}\mathbf{x}^\top.
\tag{50}
$$

$\square$

### A.8   *Acceleration in Inference*

*In this section, we analysis how our method accelerate the model in inference, both in theory and experiments.*

#### A.8.1   *FLOPs and Inference Throughput Analysis*

*Theoretically, our method which replaces LN with RMSNorm has acceleration in FLOPs, thus in throughput.*

**FLOPs**   *Floating point operations only reduces by the replacement of LN in inference. To clarify, consider a sample, with dimension $d$. We have the equation of LN and RMSNorm as follows:*

$$\text{LN}(x) = \frac{x - \mu}{\sqrt{\sigma^2 + \epsilon}}, \text{ where } \mu = \frac{1}{d}\sum_{i=1}^{d} x_i \text{ and } \frac{1}{d}\sum_{i=1}^{d}(x_i - \mu)^2. \tag{51}$$

$$\text{RMS}(x) = \frac{x}{\sqrt{\sigma_{rms}^2 + \epsilon}} \text{ where } \sigma_{rms}^2 = \frac{1}{d}\sum_{i=1}^{d} x_i^2. \tag{52}$$

*According to the formula above, we compute the operation in the Table.*

Table 3: *FLOPs calculation and computation order for 'LN' and 'RMSNorm'.*

| Operation\Equations | $\mu$ | $\sigma_{LN}^2$ | $\sigma_{RMS}^2$ | $(x-\mu)$ | $\sqrt{\sigma^2+\epsilon}$ | $\frac{\text{num}}{\text{den}}$ | scaling | bias |
|---|---|---|---|---|---|---|---|---|
| + | (d-1) | (d-1) | (d-1) | \ | \ | \ | \ | d |
| - | \ | d | \ | d | 1 | \ | \ | \ |
| \or $\times$ | 1 | 1 | 1 | \ | \ | d | d | \ |
| $\sqrt{\phantom{x}}$ | \ | \ | \ | \ | 1 | \ | \ | \ |
| $\phantom{x}_2$ | \ | d | d | \ | \ | \ | \ | \ |
| Total | d | 3d | 2d | d | accelerated | d | d | d |
| LN = $7d$ | 1st | 2nd | \ | (2nd) | 3rd | 4th | 5th | 6th |
| RMS = $4d$ | \ | \ | 1st | \ | 2nd | 3rd | \ | 4th |

**Inference Throughput**   *It should be noted that we did not specifically calculate or test the inference throughput. We believe that there will an improvement of about 10% which is consistent with the latency reduction.*

*This is because throughput not only involve the speed of model, but also the memory occupation. Our method mainly involve one centering operation and the relative bias in affine transformation in inference. Therefore, there is no significant reduction in the complexity of the model. We verified this opinion on GPT-2 and found equivalent memory usage with and without our method. Consequently, the main decisive factor for throughput is the computational speed. Thus, we can refer to the calculations and conclusions we made above regarding inference latency.*

#### A.8.2   *Inference Acceleration Experiment*

*We conduct two experiment on acceleration in inference stage. One focuses on the overall time, the other focuses on the CUDA time.*

*To be mentioned, the LN and RMSNorm we used in this experiment is implemented by our team. This means that neither LN nor RMSNorm exhibited acceleration in our tests. The reason for this is that PyTorch's implementation of LN already includes sophisticated acceleration algorithms. RMSNorm still requires optimization, although it has already been implemented.*

*We have notice the result is quite unstable. We suggest it is according to our limit on devices, the start time span of the comparison groups is relatively large, which leading to changes in computational resources, such that the result is not stable.*

**Total Inference Time Usage**   *We conducted inference-time experiments on the realistic model GPT-2, as mentioned in Section 4.4. Our method has a acceleration of 10.31% in inference.*

*We conduct 5 runs, each averaging 300 independent inference processes. We measuring the time from the first Token input to the last token output. Here we listed the quantitative time computation in the table below:*

Table 4: *Statistical results for 'LN' and 'CCWT+RMS' on GPT-2 in inference.*

| Statistic | Original (1e-6 s) | Our method (1e-6 s) | Acceleration | Average |
|---|---|---|---|---|
| Average | 15272.76 | 13749.12 | 9.98% | 9.75% |
| Trimmed Mean | 15205.37 | 13637.70 | **10.31%** | 11.25% |

*To be more specified, we list the specific time data below.*

Table 5: *Average time usage of 300 independent inference process for 'LN' and 'CCWT+RMS' on GPT-2.*

| Method | 1 | 2 | 3 | 4 | 5 | Average |
|---|---|---|---|---|---|---|
| LN | 16381.94 | 14365.73 | 14866.47 | 14782.23 | 15967.41 | 15272.76 |
| CCWT+RMS | 13817.87 | 13484.44 | 12951.35 | 14881.17 | 13610.78 | 13749.12 |
| Acceleration | 15.65% | 6.13% | 12.88% | -0.67% | 14.76% | 9.75% |

**CUDA Time Usage**   *We also conduct validation experiments on GPT-2, BERT and Bloom. Utilizing 'torch.profile', we trace the total CUDA time usage on a single A100 GPU. We conducted 10 runs with 1000 evaluations each for GPT-2, Bert and Bloom. We list the average CUDA time and statistical results (mean, variance and coefficient of variation) among 10 runs:*

Table 6: *Statistical results of CUDA time usage on 3 models between 'LN' and 'CBWC+RMSNorm'.*

| Model | Our method (s) | Original (s) | Acceleration | Mean | Variance | CV |
|---|---|---|---|---|---|---|
| BERT | 5.024 | 6.101 | 17.65% | 17.01% | 8.45% | 0.038497 |
| GPT-2 | 3.95 | 4.907 | 19.50% | 18.48% | 10.65% | 0.576306 |
| Bloom | 0.384 | 0.438 | 12.42% | 12.42% | 0.48% | 0.496713 |

*The effect of our method is quite evident, leading to a 10% to 20% in acceleration. To be noted, according to our limit on devices, the start time span of the comparison groups is relatively large, which leading to changes in computational resources, such that the result is not stable.*

*Although, we also measure the CPU time in the experiment, the acceleration is not significant, especially in Bloom. We think this is due to its overall evaluation time is short, which increases the proportion of time consumption caused by script calls. This is not what our method focuses on. However, the overall time of both CUDA and CPU enjoys an acceleration.*

A.9   *Foldable LN*

*Here, we list 11 common models and the number of LN and foldable LN. We also calculate the foldable LN after adding a centering operation after embedding layer (same as the method we applied on Pre-Norm Transformer).*

Table 7: *Number of LN, foldable LN and foldable LN with centered Embedding in 11 common models.*

| Model | Total | Foldable | Percentage | Foldable (Centered Embedding) | Percentage |
|-------|-------|----------|------------|-------------------------------|------------|
| GPT-2 | 25 | 0 | 0 | 25 | 100.00% |
| BERT | 25 | 24 | 96.00% | 25 | 100.00% |
| ViT | 25 | 0 | 0 | 25 | 100.00% |
| Phi | 25 | 0 | 0 | 25 | 100.00% |
| Phi3 | 0 | 0 | / | 0 | / |
| Qwen2 | 0 | 0 | / | 0 | / |
| T5 | 32 | 0 | 0 | 32 | 100.00% |
| OPT | 25 | 0 | 0 | 1 | 4.00% |
| BLOOM | 6 | 5 | 83.33% | 6 | 100.00% |
| Mamba2 | 0 | 0 | / | 0 | / |
| LLaMA | 0 | 0 | / | 0 | / |

*Here we can see few models do not have LN. For example, LLaMA originally uses RMSNorm. Moreover, as Pre-Norm transformer structures are widely used, many models requires a additonal centering operation after embedding layer.*

### A.10 EMPIRICAL EXPERIMENTS

All of our experiments are conducted on one 3090Ti.

#### A.10.1 ABLATION EXPERIMENT OF CENTERING OPERATION

The depth of MLP varies from 6, 15, 35 on CIFAR-10, with width of 256 and 512 (the result in Section 5.1). We introduce residual structure to help converge for deeper MLP with depth of 65 and 100 on MNIST, with width of 512. We train the model with learning rate 0.01 and batch size 256. We train all the model for 175 epochs. The results of the experiments are in the following tables.

Table 8: Accuracy results for MLP in classification task on CIFAR-10.

| Depth | Width | Norm | Best Test Acc | Last Test Acc | Last Train Acc |
|---|---|---|---|---|---|
| 6 | 256 | LN | 42.85% | 41.05% | 100.00% |
| 6 | 256 | RMSNorm | 42.07% | 39.97% | 100.00% |
| 6 | 512 | LN | 44.10% | 43.38% | 100.00% |
| 6 | 512 | RMSNorm | 42.08% | 41.71% | 100.00% |
| 15 | 256 | LN | 42.47% | 41.80% | 99.99% |
| 15 | 256 | RMSNorm | 41.47% | 40.35% | 100.00% |
| 15 | 512 | LN | 44.62% | 44.52% | 100.00% |
| 15 | 512 | RMSNorm | 43.92% | 43.76% | 100.00% |
| 35 | 256 | LN | 43.00% | 41.00% | 99.80% |
| 35 | 256 | RMSNorm | 42.20% | 39.59% | 99.87% |
| 35 | 512 | LN | 45.05% | 43.96% | 99.98% |
| 35 | 512 | RMSNorm | 42.87% | 42.12% | 99.99% |

Table 9: Accuracy results for MLP with residual structure in classification task on MNIST.

| depth | norm | best test acc | last test acc | last train acc |
|---|---|---|---|---|
| 65 | LN | 98.10% | 98.03% | 99.58% |
| 65 | RMSNorm | 98.04% | 97.97% | 99.50% |
| 100 | LN | 98.07% | 98.02% | 99.61% |
| 100 | RMSNorm | 98.03% | 97.96% | 99.54% |

### A.10.2  *Empirical Experiment on SWIN*

*For the Imagenet100, we select 100 classes from Imagenet1k (Deng et al., 2009) according to the given classes in (Tian et al., 2019).*

*We chose SWIN-T for this experiment and train on a single 3090. The time result in Table 2 is averaged for 40 epochs. For each epoch, we record the average time among all the batches.*

*Here we list the train accuracy for each seed under the three method and their mean, variance and coefficient of variation. Our method is more stable than the other two.*

Table 10: *Train accuracy for different random seed for SWIN on Imagenet100.*

| Seed\Method | LN | CBWC+RMS | RMS |
|---|---|---|---|
| 128 | 95.07157898 | 94.37104797 | 95.03309631 |
| 42 | 95.18504333 | 94.56147003 | 95.08243561 |
| 2 | 95.1505127 | 94.69861603 | 95.12683105 |
| 1 | 94.55850983 | 94.26053619 | 94.43616486 |
| Mean | 94.99141121 | 94.47291756 | 94.91963196 |
| Variance | 0.253296842 | 0.169015233 | 0.281092108 |
| CV | 0.002666524 | 0.001789034 | 0.00296137 |

### A.10.3  VERIFICATION EXPERIMENT FOR CONTINUE LEARNING

We conduct classification task on CIFAR-10. The model has a depth of 6 and a width of 256. We train the model for 40 epochs with learning rate 0.01 and batch size 256 under one seed.

