# OpenReview forum: "Enjoy Your Layer Normalization with the Computation Efficiency of RMSNorm"
_ICLR.cc/2025/Conference — Submitted to ICLR 2025_

### Official Review · Reviewer_DErw · 2024-11-03

**Soundness:** 3
**Presentation:** 2
**Contribution:** 2
**Rating:** 5
**Confidence:** 4

**Summary:**

In this work the authors context the computational complexity led by layer normalization, proposing to use instead RMS Normalization. The equivalence between the two normalization is established by ensuring that the signal at the output of a layer has average zero, which can be achieved by imposing specific constraints to the input of such layer, and to the parameter's distribution. The authors test the performance achieved with their proposed approach on simple image classification tasks (MNIST and CIFAR-10) and on text translation and classification.

**Strengths:**

- The objective is clear, the writing in most of the parts is also clear.
- The approach is in general reasonable.

**Weaknesses:**

- The approach has some implicits, like that there should be no non-linearity between the convolution/fully connected layer and the layer norm.
- The validation is performed on relatively small-scale datasets.
- There is no quantitative time computation reduction for both training and inference time.

**Questions:**

- How is the approach performing on larger datasets (like ImageNet-1k) on realistic architectures like SWIN?
- What is the real wall-clock time in employing the proposed CWBC+RMSNorm compared to LayerNorm?
- What are the cases this approach is applicable (ie. what are the specific constraints to the architecture to make this solution usable)?
- Is there any major attention to pay for hyper-parameters fine-tuning? Or the approach is completely insensitive to large variation of hyper-parameters like learning rate and weight decay?
- Gradient-wise, is CWBC+RMSNorm equivalent to LayerNorm?

---

> ### Author Response · Authors · 2024-11-15
>
> **Weakness1:** 'The approach has some implicits, like that there should be no non-linearity between the convolution/fully connected layer and the layer norm.'
>
> **Answer:** Thank you for your mentioning. Indeed, the convolution/fully connected layer itself is entirely linear, while the centering operation in LayerNorm is a linear transformation, represented as a hyper-plane projection. However, the scaling operation, represented as a hyper-sphere projection, includes non-linearity, which is proven in this paper[1]. Yet, our method (CCWT/CBWC/CCC) does not introduce any additional nonlinearity,  as we retain the inherent scaling operation in LayerNorm (or what we called RMSNorm).
>
> [1] Ni Y, Guo Y, Jia J, et al. On the Nonlinearity of Layer Normalization[J]. arXiv preprint arXiv:2406.01255, 2024.
>
> **Question3:** 'What are the cases this approach is applicable (ie. what are the specific constraints to the architecture to make this solution usable)?'
>
> **Answer:**  Thank you for your inquiry. We are not entirely certain that we have understood your question correctly.
>
> For linear layers, the constraint is described in Eqn6, and the transformation in CCWT is detailed in Eqn8. For other regulable modules such as convolutional layers, recurrent layers, and self-attention mechanisms, the proofs and constraints are provided in Appendix A.3. Below, I've added the transformations in CCWT for these layers, which will also be included in the revised version of our paper:
>
> Recurrent Neural Network: $W^v = \Phi(W^v)=(I-\frac{1}{m_v}\pmb 1_{m_v}^\top\pmb 1_{m_v})W^v,\ W^h = \Phi(W^h)=(I-\frac{1}{m_h}\pmb 1_{m_h}^\top\pmb 1_{m_h})W^h$
>
> Convolutional Layer: $W = \Phi(W)=(I-\frac{1}{h\times w}\pmb 1_{h\times w}^\top\pmb 1_{h\times w})W$
>
> Self-attention: $W^v = \Phi(W^v)=(I-\frac{1}{m_v}\pmb 1_{m_v}^\top\pmb 1_{m_v})W^v$
>
> Regarding the algorithm to apply these constraints to any architectures or models, we have discussed in Section 4.3 and Section 4.4.
>
> We have developed a tool that automates the application of our method for validation purposes. This tool will assist in implementing the transformations and constraints across different types of layers and models efficiently.
>
> **Question5:** 'Gradient-wise, is CWBC+RMSNorm equivalent to LayerNorm?'
>
> **Answer:** Thank you for your question. Yes. Under the condition that CBWC is directly before RMSNorm without any dropout layers in between, ‘CBWC+RMSNorm’ has the same learnable parameters and performs the same forward pass as Layer Normalization. They are equivalent in terms of gradients, as we have proved in Appendix A.7.
>
> It is important to note that this equivalence holds when CBWC is placed directly before RMSNorm without any dropout layers in between. If a dropout layer is introduced, it can disrupt the zero-mean property of the output of CBWC during training, which can, in turn, cause the gradients—and consequently the results—of ‘CBWC+RMSNorm’ to differ from those of LayerNorm.
>
> Additionally, we sincerely apologize for the misspelling of 'CBWC' in our paper; it should be 'CBWC' instead of 'CWBC'. We will correct this error in the revised version of our paper.

---

> ### Author Response · Authors · 2024-11-18
>
> **Question4:** 'Is there any major attention to pay for hyper-parameters fine-tuning? Or the approach is completely insensitive to large variation of hyper-parameters like learning rate and weight decay?'
>
> **Answer:** Thank you for your inquiry. This is a good question. If you employ 'CBWC+RMSNorm' when CBWC is directly before RMSNorm, then, as we demonstrated in Proposition 3, there is no difference between our method and a model based on original LayerNorm, whether in terms of hyper-parameter tuning or performance and outcomes.
>
> However, if there are modules interposed between CBWC and RMSNorm (for example dropout) that disrupt the zero-mean property, differences will emerge thus may need different hyper-parameter. Under such circumstances, as we stated in our response to Reviewer z8VX, we recommend using 'CCWT+RMSNorm' exclusively during the validation stage for fine-tuned models. This approach ensures the greatest consistency with the original model and effectively accelerates the process, eliminate the potential instability and addition attention for hyper-parameters.

---

> > ### Comment · Reviewer_DErw · 2024-11-25
> >
> > I thank the authors for addressing some of my concerns. Some of my questions remain unaddressed - although I see the value of this work, I still believe some polishing for the method/writing and more comparisons are required; hence, I maintain my original score.

---

> > > ### Author Response · Authors · 2024-11-28
> > >
> > > Thank you for your comment, and we apologize for the late response. Regarding your other questions, we have added some experiments and descriptions which can be found in the replies and the latest version of the paper, hoping it will be helpful to you.

---

> ### Author Response · Authors · 2024-11-28
>
> **Weakness2:** 'The validation is performed on relatively small-scale datasets.' **&Weakness3:** 'There is no quantitative time computation reduction for both training and inference time.' **& Question1: ** 'How is the approach performing on larger datasets (like ImageNet-1k) on realistic architectures like SWIN?' **& Question2: **'What is the real wall-clock time in employing the proposed CWBC+RMSNorm compared to LayerNorm?'
>
> **Answer:** Thank you for your question. Sorry for taking it so long to reply.
>
> For inference time evaluation, we have conducted experiments on GPT-2, BERT, and Bloom, as we mentioned in our reply to Reviewer z8VX. For training stage, we also add a few experiments on training SWIN-T (on a single 3090) on Imagenet100. We add more details into our revised-version of paper.

---

> ### Author Response · Authors · 2024-11-30
>
> Dear Reviewer DErw,
>
> Grateful thanks for your feedback and the question you've raised, which are valuable and comprehensive. We apologize for the lengthy time it took to conduct experiments on SWIN, which resulted in delayed responses. We have revised our paper and hope that the results in our experimental section can address your concerns. As the end of the discussion period is approaching, we kindly encourage you to share new feedback with us. We highly value your insights. We would be appreciated if the additional experiments we've included in our revised paper can address your concerns and help with your assessment.
>
> Thank you again for your comments.
>
> Sincerely,
>
> The Authors.

---

### Official Review · Reviewer_T7BL · 2024-11-04

**Soundness:** 3
**Presentation:** 3
**Contribution:** 2
**Rating:** 5
**Confidence:** 4

**Summary:**

This paper introduces a novel approach to enhance the computational efficiency of Layer Normalization (LN) while retaining its theoretical advantages. By defining a column centering constraint, the authors demonstrate how to achieve the same output as LN without the centering operation, thereby reducing computation costs during inference. The paper also presents techniques such as Column Centered Weight Transformation (CCWT) and Column Based Weight Centering (CBWC) to ensure linear modules maintain centering during training. Experimental results indicate that these methods perform well across various pre-trained large models, effectively improving computational efficiency.

**Strengths:**

+ The paper presents a clear and rigorous theoretical framework for removing the centering operation from Layer Normalization (LN), establishing the conditions under which this is feasible. The introduction of column-centered constraints provides a solid foundation for justifying the equivalence of the proposed method (RMSNorm) to traditional LN, which could inspire further research into optimization techniques in neural network training.

+ The authors effectively demonstrate the applicability of their method to various pre-trained large language models (LLMs) and vision-language models (VLMs). This not only showcases the versatility of their approach but also addresses a pressing issue in the field: the computational inefficiency of LN during inference. The direct application of their method in real-world scenarios adds significant value.

+ The paper includes experiments that validate the effectiveness of the proposed RMSNorm combined with column-based weight centering (CBWC) during both training and inference. The results indicate that the new method can achieve comparable performance to LN while significantly reducing computational costs, providing empirical evidence to support the theoretical claims made throughout the paper.

**Weaknesses:**

- While the paper introduces RMSNorm, the novelty of the method may be perceived as incremental rather than groundbreaking. The authors reference some similar normalization techniques (e.g., Batch Normalization, Group Normalization) but could strengthen their contribution by providing a more detailed comparison of these methods in terms of performance metrics and theoretical underpinnings. Specifically, addressing how RMSNorm differentiates itself from existing approaches in both practice and theory could enhance the perceived novelty.

- The experimental evaluation is primarily conducted on a limited set of benchmarks. Testing RMSNorm on large language models, such as those based on the LLaMA architecture, would be beneficial. This would help assess the method's effectiveness in the context of contemporary model architectures and could reveal how RMSNorm performs in scenarios typical of large-scale language tasks, including language generation and understanding.

- The paper would benefit from more extensive ablation studies that explore the impact of different components of the proposed method. For example, examining how variations in the column-centered constraints influence performance could offer insights into the critical elements that contribute to the effectiveness of RMSNorm. This would not only strengthen the paper's claims but also guide future researchers in understanding the design choices involved.

**Questions:**

What specific performance metrics differentiate RMSNorm from existing methods like Batch Normalization and Group Normalization? Can the authors provide a clearer comparison?

Will the authors consider testing RMSNorm on large language models, such as LLaMA, to assess its effectiveness in large-scale language tasks?

Could the authors conduct more ablation studies to clarify how variations in column-centered constraints affect RMSNorm's performance?

---

> ### Author Response · Authors · 2024-11-15
>
> **Weakness1:** '...The authors reference some similar normalization techniques (e.g., Batch Normalization, Group Normalization) but could strengthen their contribution by providing a more detailed comparison of these methods in terms of performance metrics and theoretical underpinnings. Specifically, addressing how RMSNorm differentiates itself from existing approaches in both practice and theory could enhance the perceived novelty.' **& Questions1:** 'What specific performance metrics differentiate RMSNorm from existing methods like Batch Normalization and Group Normalization? Can the authors provide a clearer comparison?'
>
> **Answer:** Thank you for your comment. We are not entirely certain that we have understood your question correctly. To clarify, the objective of our paper is to identify a method that accelerates LayerNorm by applying a transformation to the linear module preceding it, which allows for the replacement of LayerNorm with RMSNorm while maintaining equivalent results. Therefore, the differences in accuracy or training stability between normalization methods are outside the scope of our considerations; our primary focus is on the efficiency of LayerNorm and RMSNorm. Here, we outline the differences between these normalization techniques in terms of their definitions.
>
> Normalization is primarily about ensuring training stability and aiding optimization by selecting statistical measures that remap the sample distribution to achieve zero-mean and unit-variance. The distinctions between BatchNorm[1], LayerNorm[2], GroupNorm[3], and RMSNorm[4] lie in their choice of statistical measures, leading to different outcomes and impacts on training. BatchNorm involves computing the mean and variance of a group of samples within a batch. LayerNorm normalizes all elements within each sample. GroupNorm is an extension of LayerNorm, grouping the channels of each sample and normalizing the elements in each group respectively. RMSNorm is essentially a scaling operation of LayerNorm, aiming to approach unit-variance by dividing by the sample's second-order distance, neglecting the mean. To our best knowledge, each has its own advantages in terms of performance due to their different abilities in optimization, generalization, and representation capacity.
>
> In current large-scale models, due to the variable length of sequences, LayerNorm is commonly used as a fundamental component of architectures like Transformer or Mamba. However, some models opt for RMSNorm to reduce the computational overhead of LayerNorm. (RMSNorm is a version of LayerNorm that omits the centering operation. Replacing LayerNorm with RMSNorm can significantly reduce computational costs. With appropriate optimizations, RMSNorm can be faster than LayerNorm.) For instance, the original LLaMA model uses RMSNorm instead of LayerNorm in its transformer.
>
> Therefore, in our paper, we do not comment on whether RMSNorm or any other normalization method performs better. We focus on how RMSNorm can accelerate the model and how to minimize any performance reduction that may result from replacing LayerNorm.
>
> [1] Ioffe S. Batch normalization: Accelerating deep network training by reducing internal covariate shift[J]. arXiv preprint arXiv:1502.03167, 2015.
>
> [2] Ba J L. Layer normalization[J]. arXiv preprint arXiv:1607.06450, 2016.
>
> [3] Wu Y, He K. Group normalization[C]//Proceedings of the European conference on computer vision (ECCV). 2018: 3-19.
>
> [4] Zhang B, Sennrich R. Root mean square layer normalization[J]. Advances in Neural Information Processing Systems, 2019, 32.

---

> ### Author Response · Authors · 2024-11-15
>
> **Weakness3:** 'The paper would benefit from more extensive ablation studies that explore the impact of different components of the proposed method. ...' **& Question3:** 'Could the authors conduct more ablation studies to clarify how variations in column-centered constraints affect RMSNorm's performance?'
>
> **Answer:** Thank you for your question. We are not entirely certain that we have understood your question correctly.
>
> It is important to note that our paper does not aim to improve the performance of RMSNorm. CCC acts as an alternative to the centering operation in LayerNorm, which RMSNorm neglects. Our focus is on the efficiency gains that can be achieved by using RMSNorm in place of LayerNorm, particularly in scenarios where computational efficiency is a concern.
>
> Different forms of Column Centered Constraints (CCC) vary depending on the type of linear modules they are associated with—such as linear layers, convolutional layers, or recurrent layers—but they all serve the same purpose within the models. CCC acts as a centering function equivalent to that of LayerNorm, which means that the subsequent LayerNorm has a redundant centering operation, allowing it to be replaced with RMSNorm safely. Thus, CCC is a necessary precondition for using RMSNorm while maintaining the same performance as LayerNorm. Without CCC, RMSNorm and LayerNorm will not be equivalent.
>
> If your question pertains to how CCC can improve performance in a model that originally contains RMSNorm, we have implied our conclusion in Section 5.1 (CCC serves as a centering operation equivalent to LayerNorm when there is no dropout). Even with dropout that disturbs the zero-mean, CCC can still have a certain effect, as shown in Section 5.3.

---

> ### Author Response · Authors · 2024-11-26
>
> **Weakness2:** 'The experimental evaluation is primarily conducted on a limited set of benchmarks. Testing RMSNorm on large language models, such as those based on the LLaMA architecture, would be beneficial. This would help assess the method's effectiveness in the context of contemporary model architectures and could reveal how RMSNorm performs in scenarios typical of large-scale language tasks, including language generation and understanding.' **& Question2：**'Will the authors consider testing RMSNorm on large language models, such as LLaMA, to assess its effectiveness in large-scale language tasks?'
>
> **Answer:** Thank you for your inquiry. We are not certain whether we completely understand your question. In validation, our method will not affect the result, due to theoretical equivalence between 'CCWT+RMS' and 'LN'. Therefore, changing LN into RMSNorm after using CCWT will not affect the performance of the original model.
>
> In the training stage, under some circumstances, using 'CBWC+RMSNorm' with dropout and residual connections between them can lead to inconsistencies. This means that the results of our method will differ from vanilla results. Yet, most large models still choose to use LN instead of RMSNorm. For some extent, RMSNorm itself neglects the effect of the centering operation (as we discussed in Section 5.1), which can lead to training and optimization problems. This can also be found out in the empirical experiments in Sections 5.2 and 5.3. We believe that the fact is more complex than this, involving training dynamics, and cannot be proven solely through empirical experiments. Hence, simply evaluating the performance of RMSNorm in large language models (LLMs) is not our aim. Here, we consider CBWC as a precondition for changing LN into RMSNorm, with the ultimate goal of accelerating LN. Our goal in this paper is to improve the speed of LN through this method while ensuring that the effect remains unchanged or the degree of deterioration is minimal. We are not focused on discussing the capabilities of RMSNorm itself.
>
> It is worth mentioning that in certain tasks RMSNorm does achieve better results than LN. We find that under models such as LLaMA and Mamba, RMSNorm was selected in the original paper instead of LN, in order to achieve faster speed or better results. We believe that this is not within the scope of our paper's discussion.

---

> ### Author Response · Authors · 2024-11-30
>
> Dear Reviewer T7BL,
>
> Thank you very much for your valuable opinions and the questions you've raised. We have responded to your inquiries and made revisions to our paper. If you have any further questions or doubts, or any dissatisfaction to our response, we would be glad to discussion with you. We kindly encourage you to provide additional feedback which is helpful for us to clarify and enhance the quality of our article, before the discussion phase comes to an end. If you feel that our responses have addressed your concerns, we would appreciate it if you could re-evaluating our paper and reconsider your score.
>
> Thank you again for your comments.
>
> Sincerely,
>
> The Authors.

---

### Official Review · Reviewer_z8VX · 2024-11-04

**Soundness:** 3
**Presentation:** 3
**Contribution:** 2
**Rating:** 5
**Confidence:** 3

**Summary:**

The paper introduces a method to balance the theoretical benefits of Layer Normalization (LN) with the computational efficiency of RMSNorm. The authors propose Column Centered Weight Transformation (CCWT) and Column Based Weight Centering (CBWC) as techniques to achieve the effects of LN without the need for centering, which reduces computational overhead. This approach enables the usage of RMSNorm in place of LN for various deep learning models, particularly large language models (LLMs) and vision-language models (VLMs), without compromising performance. The paper demonstrates both theoretical equivalence and empirical results to support the proposed method's efficiency during inference and training.

**Strengths:**

(1) The approach reduces inference time by removing the need for centering in LN. This can be beneficial for LLMs and VLMs, where inference latency and computational resources are critical.
(2) The authors provide rigorous definitions and proofs to justify the proposed transformations, ensuring that the new method is theoretically sound.
(3) The CCWT and CBWC techniques can be applied across various pre-trained models, enabling easy adoption in different architectures with LN.
Experimental Validation: The experiments confirm that the method maintains performance on translation and classification tasks while enhancing efficiency, showing promise in practical applications.

**Weaknesses:**

(1) Limited Evaluations in Experiments: Although the method is broadly applicable, the empirical tests are limited to a few models. Further experimentation on a wider variety of large language models, especially decoder-only models on text-generation tasks, could provide a stronger justification for generalizability.
(2) Potential Training Instabilities: The method, though theoretically equivalent, could exhibit slight instabilities during training, as shown in the empirical results, especially when Dropout Layers disrupt the zero-mean property.
(3) Lack of Inference-Time Statistics: While the paper focuses heavily on inference efficiency, it lacks extensive analysis on the comparisons between the proposed method and baseline transformers, especially regarding realistic inference statistics such as inference latency, FLOPs and inference throughput.

**Questions:**

See the weaknesses.

---

> ### Author Response · Authors · 2024-11-18
>
> **Weakness2:** 'Potential Training Instabilities: The method, though theoretically equivalent, could exhibit slight instabilities during training, as shown in the empirical results, especially when Dropout Layers disrupt the zero-mean property.'
>
> **Answer:** Thank you for bringing this to our attention. It is important to note that if you apply 'CBWC+RMSNorm' with CBWC directly preceding RMSNorm, the theoretical equivalence suggests that both performance and outcomes will be identical. Therefore, there should be no potential training instability.
>
> However, we agree that there are potential training instabilities in the case you mentioned, where dropout is placed between CBWC and RMSNorm. This is because dropout disrupts the zero-mean property and diminishes the effectiveness of CBWC in achieving its intended outcome (or what the original centering operation is intended to achieve). Nonetheless, linear modules within the 'Regulable Module' are still subject to the constraints imposed by CBWC. Therefore, we propose 'CBWC+RMSNorm' as an alternative training approach, that can lead to shorter training times (especially for cases with long sequences) or better performance (in certain specific scenarios).
>
> In the cases you mentioned, dropout layers function as element-wise scaling during evaluation and thus do not impact the zero-mean property. If efficiency during the validation stage is a priority, you can also apply our method by employing LayerNorm for training and then switching to 'CCWT+RMSNorm' for validation.  This approach eliminates concerns about training instabilities and allows us to enjoy on the efficiency of RMSNorm. Thus, 'CBWC+RMSNorm' is designed for situations where 'CCWT+RMSNorm' is not suitable, such as when there are additional intermediate modules that must be considered during the validation stage and cannot be ignored.

---

> ### Author Response · Authors · 2024-11-26
>
> **Weakness3:** 'Lack of Inference-Time Statistics: While the paper focuses heavily on inference efficiency, it lacks extensive analysis on the comparisons between the proposed method and baseline transformers, especially regarding realistic inference statistics such as inference latency, FLOPs and inference throughput.'
>
> **Answer:** Thank you for pointing out this.
>
> 1. Inferency latency:We conducted inference-time experiments on the realistic model GPT-2, as detailed in Section 4.4. Our method has a acceleration of 10.31% in inference. This result is derived from five runs, each averaging 300 independent inference processes.  Here we listed the quantitative time computation in the table below (further details will be included in the appendix of the revised paper):
>
> |    Statistic | Original (1e-6 s) | Our method (1e-6 s) | Acceleration | Average |
> | -----------: | :---------------: | :-----------------: | :----------: | :-----: |
> |      Average |     15272.76      |      13749.12       |    9.98%     |  9.75%  |
> | Trimmed Mean |     15205.37      |      13637.70       |  **10.31%**  | 11.25%  |
>
> We also conduct more validation experiments on GPT-2, BERT and Bloom. Utilizing `torch.profile`, we trace the total CUDA time usage on a single A100 GPU. We conducted 10 runs with 1000 evaluations each for GPT-2, Bert and Bloom. Here, we list the average CUDA time and statistical results among 10 runs:
>
> | Model | Our method (s) | Original (s) | Acceleration | Average  | Variance | Coefficient of Variation |
> | ----: | :------------: | :----------: | :----------: | :------: | :------: | :----------------------: |
> |  BERT |     5.024      |    6.101     |   17.6490%   | 17.0078% | 8.4480%  |       0.038496672        |
> | GPT-2 |     3.950      |    4.907     |   19.5036%   | 18.4841% | 10.6525% |       0.576305538        |
> | Bloom |     0.384      |    0.438     |   12.4195%   | 12.4185% | 0.4781%  |        0.49671261        |
>
> According to this table, we can notice an acceleration of 10%-20% in efficiency in CUDA time. To be noted, according to our limit on devices, the start time span of the comparison groups is relatively large, which leading to changes in computational resources, such that the result is not stable. But the effect of our method is quite evident. We will include more detailed data in the revised paper.
>
> To be mentioned, the LN and RMSNorm we used in this experiment is implemented by our team. This means that neither LN nor RMSNorm exhibited acceleration in our tests. The reason for this is that PyTorch's implementation of LN already includes sophisticated acceleration algorithms. However, RMSNorm still requires optimization. We have noticed that RMSNorm has been added to PyTorch(https://github.com/pytorch/pytorch/issues/128713).  We are optimistic that more efficient acceleration methods for RMSNorm will be developed soon.
>
> 2. FLOPs: Floating point operations only reduces by the replacement of LN in inference.
>
> To clarify, here we list the FLOPs of LN and RMSNorm below: Consider a sample, with dimension $d$. We compute the operation counts separately as follows:
>
> -   $\text{LN}(x)=\frac{x-\mu}{\sqrt{\sigma^2+\epsilon}}$, where $\mu = \frac{1}{d}\sum_{i=1}^dx_i$ and $\frac{1}{d}\sum_{i=1}^d(x_i - \mu)^2$.
>
> -   $\text{RMS}(x)=\frac{x}{\sqrt{\sigma_{rms}^2+\epsilon}}$, where $\sigma_{rms}^2 = \frac{1}{d}\sum_{i=1}^d x_i ^2$.
>
> |     Operation | $\mu $ | $\sigma_{LN}^2$ | $\sigma_{RMS}^2$ | $(x-\mu)$ | $\sqrt{\sigma^2 + \epsilon}$ | $\frac{\text{num}}{\text{den}}$ | Scaling | Bias |
> | ------------: | :----: | :-------------: | :--------------: | :-------: | :--------------------------: | :-----------------------------: | :-----: | :--: |
> |             + | (d-1)  |      (d-1)      |      (d-1)       |     \     |              \               |                \                |    \    |  d   |
> |             - |   \    |        d        |        \         |     d     |              1               |                \                |    \    |  \   |
> | \ or $\times$ |   1    |        1        |        1         |     \     |              \               |                d                |    d    |  \   |
> |     $\sqrt{}$ |   \    |        \        |        \         |     \     |              1               |                \                |    \    |  \   |
> |          $^2$ |   \    |        d        |        d         |     \     |              \               |                \                |    \    |  \   |
> |         Total |   d    |       3d        |        2d        |     d     |         Accelerated          |                d                |    d    |  d   |
> |     LN = $7d$ |  1st   |       2nd       |        \         | (in 2nd)  |             3rd              |               4th               |   5th   | 6th  |
> |    RMS = $4d$ |   \    |        \        |       1st        |     \     |             2nd              |               3rd               |    \    | 4th  |

---

> ### Author Response · Authors · 2024-11-26
>
> In general, according to this table, we have $7d$ operations in LN and $4d$ operations in RMSNorm.
>
> Furthermore, in the calculation of LayerNorm, all calculations must be done after computing $\mu$. However, in the case of RMSNorm, there is no need to calculate $μ$ (which is a computational bottleneck). The calculation order is detailed in the table above. Therefore, our method offers advantages in both computational complexity and parallel computing efficiency.
>
> 3. Inference throughput: It should be noted that we did not specifically calculate or test the inference throughput. We believe that there will an improvement of about 10% which is consistent with the latency reduction.
>
> This is because throughput not only involve the speed of model, but also the memory occupation. Our method mainly involve one centering operation and the relative bias in affine transformation in inference. Therefore, there is no significant reduction in the complexity of the model. We verified this opinion on GPT-2 and found equivalent memory usage with and without our method.  Consequently, the main decisive factor for throughput is the computational speed. Thus, we can refer to the calculations and conclusions we made above regarding inference latency.

---

> ### Author Response · Authors · 2024-11-28
>
> **Weakness1:**  'Limited Evaluations in Experiments: Although the method is broadly applicable, the empirical tests are limited to a few models. Further experimentation on a wider variety of large language models, especially decoder-only models on text-generation tasks, could provide a stronger justification for generalizability. '
>
> **Answer:** Thank you for your comment. Sorry for replying so late. We add more experiments on SWIN for Imagenet100. We add our result into revised version of our paper.

---

> ### Author Response · Authors · 2024-11-30
>
> Dear reviewer z8VX,
>
> We sincerely thank you for your valuable feedback. We have replied and revised the paper according to the issues and suggestions you raise. As the end of the discussion phase is coming, we kindly encourage you to share new feedback with us. We hope our responses have addressed your concerns and we would be appreciate if you can update your views on our paper, as well as the score. We highly value your opinion. If there is any remaining questions or points of clarification, please let us know. We would be glad to address your concerns.
>
> Thank you again for your comments.
>
> Sincerely,
>
> The Authors.

---

### Meta-Review · Area_Chair_g1bS · 2024-12-19

**Metareview:**

The paper presents a method to optimize Layer Normalization (LN) by leveraging the computational efficiency of RMSNorm without compromising the theoretical benefits of LN, using techniques such as Column Centered Weight Transformation (CCWT) and Column Based Weight Centering (CBWC).

The reviewers have expressed common concerns regarding the limited empirical evaluations, perceived incremental novelty, and lack of clarity on inference-time statistics, wall-clock time savings, applicability constraints, and sensitivity to hyper-parameters.
Despite the authors' efforts in the rebuttal, some concerns remain unresolved. Therefore, the final consensus of negative ratings lead to a rejection for this submission.

**Additional Comments On Reviewer Discussion:**

Several concerns have been raised by the reviewers. Reviewer z8VX points out that the empirical evaluations are limited and that there is a lack of extensive analysis on inference-time statistics, such as latency, FLOPs, and throughput. Reviewer T7BL suggests that the method's novelty may be perceived as incremental and recommends broader empirical evaluations, including on contemporary large language models. Reviewer DErw questions the implicit assumptions in the approach and calls for more clarity on the wall-clock time savings, applicability constraints, and sensitivity to hyper-parameters.

The authors respond to the reviews by clarifying their method's theoretical equivalence to LN when using CBWC and RMSNorm, addressing concerns about training instabilities, and providing extensive inference-time statistics to demonstrate efficiency gains. Despite the authors' efforts in the rebuttal, some concerns remain unresolved and reviewers tend to keep the original negative scores.

---

### Decision · Program_Chairs · 2025-01-22

Reject